

# The Stratospheric Water and Ozone Satellite Homogenized (SWOOSH) database: A long-term database for climate studies

Sean M. Davis[1,2], Karen H. Rosenlof[1], Birgit Hassler[1,2], Dale F. Hurst[1,2], William G. Read[3],
5  Holger Vömel[4], Henry Selkirk[5,6], Masatomo Fujiwara[7], Robert Damadeo[8]

[1]NOAA Earth Systems Research Laboratory (ESRL), Boulder, CO
[2]Cooperative Institute for Research in Environmental Sciences (CIRES), University of Colorado at Boulder, Boulder, CO
[3]Jet Propulsion Laboratory, California Institute of Technology, Pasadena, CA
10  [4]National Center for Atmospheric Research, Boulder, CO
[5]NASA Goddard Space Flight Center, Greenbelt, MD
[6]Universities Space Research Association, Columbia MD
[7]Hokkaido University, Sapporo, Japan
[8]NASA Langley Research Center, Hampton, VA

15  *Correspondence to*: Sean Davis (sean.m.davis@noaa.gov)



**Abstract.** In this paper, we describe the construction of the Stratospheric Water and Ozone Satellite Homogenized (SWOOSH) database, which includes vertically resolved ozone and water vapor data from limb profiling satellite instruments operating since the 1980's. SWOOSH includes both individual satellite source data as well as a merged data product. A key aspect of the merged product is that the source records are homogenized to account for inter-satellite biases and to minimize artificial jumps in the record. We describe the SWOOSH homogenization process, which involves adjusting the satellite data records to a "reference" satellite using coincident observations during time periods of instrument overlap. The reference satellite is chosen based on the best agreement with independent balloon-based sounding measurements, with the goal of producing a long-term data record that is both homogeneous and accurate. This paper details the choice of reference measurements, homogenization, and gridding process involved in the construction of the combined SWOOSH product, and also presents the ancillary information stored in SWOOSH that can be used in future studies of water vapor and ozone variability. Furthermore, a discussion of uncertainties in the combined SWOOSH record is presented, and examples of the SWOOSH record are provided to illustrate its use for studies of ozone and water vapor variability on interannual to decadal time scales. The version 2.5 SWOOSH data are publicly available at https://data.noaa.gov/dataset/stratospheric-water-and-ozone-satellite-homogenized-swoosh-data-set.

## 1 Introduction

Ozone ($O_3$) and water vapor (WV) are key for determining the temperature structure and radiative balance in the stratosphere. Changes in the concentrations of these gases affect the global-mean top-of-atmosphere energy budget, surface UV radiation, and even tropospheric circulation patterns. To first order, climate impacts from changes in ozone concentration vary with the total column amount (e.g., surface UV radiation), however, other impacts are highly sensitive to changes in the vertical distribution. In particular, several studies have shown a high sensitivity of radiative forcing and local temperature structure from $O_3$/WV changes in the upper troposphere and lower stratosphere (UTLS) region (Forster and Shine, 1999; Forster et al., 2007; Maycock et al., 2011; Solomon et al., 2010).

Despite their chemical and radiative importance in the stratosphere, there have been relatively few attempts at constructing long-term data records of $O_3$ and WV based on vertically resolved satellite limb-based observations of these species. To aid in the study of variability and change in water vapor and ozone in the upper troposphere to stratosphere region, we have constructed the Stratospheric Water Vapor and Ozone Satellite Homogenized (SWOOSH) data set. SWOOSH is a global long-term vertically resolved gridded database of satellite $O_3$/WV



measurements that has been designed with the goal of accurately reproducing the monthly average variability present in the underlying data. SWOOSH can be used as input to global models to test sensitivity to changes in ozone or water vapor, as well as for comparison with model output. In this paper, we describe the construction of the database, which includes new data filtering and homogenization algorithms.

Although several efforts have been made to combine the overlapping satellite ozone measurements that began in 1978 with the Total Ozone Mapping Spectrometer (TOMS) on Nimbus-7 (Hudson, 2012; Cionni et al., 2011; Hassler et al., 2008; Randel and Wu, 2007; McLinden et al., 2009), to date there have been few attempts to create a homogenized satellite record of vertically resolved water vapor (Hegglin et al., 2013; Froidevaux et al., 2015). This is no doubt partly due to gaps in the satellite data records, and also due to well-documented

disparities in satellite (and other) measurements of water vapor (e.g., Kley et al., 2000).

Satellite vertical profile measurements of ozone and WV date back to Stratospheric Aerosol and Gas Experiment (SAGE I, ozone only, 1979-1981) and Limb Infrared Monitor of the Stratosphere (LIMS, 1978-1979; Gille and Russell, 1984; Remsberg et al., 1984), but unfortunately these records do not overlap with subsequent satellite datasets. Continuous coverage of ozone/WV vertical profiles from satellites begins only

with the SAGE II instrument in October 1984. Since then, several other NASA satellite $O_3$/WV sounders have been launched; these include the Upper Atmospheric Research Satellite Halogen Occultation Experiment (UARS HALOE, 1991-2005), the UARS Microwave Limb Sounder (MLS), the SAGE III (2001-2006), and the Earth Observing System (EOS) Aura MLS (2004 -). Since 2002 several satellite instruments measuring ozone and water vapor have also been launched by the European and Canadian space agencies (e.g., ACE-FTS, Odin-

SMR, Odin-OSIRIS, Envisat-MIPAS, and Envisat-SCIAMACHY).

The aforementioned NASA satellite ozone profile measurements have been shown to be in good agreement (~5-10%) between about 100 hPa and 1 hPa (Cunnold et al., 1996; Froidevaux et al., 2008; Jiang et al., 2007a; Livesey et al., 2008; Nazaryan and McCormick, 2005; Randall, 2003; Wang et al., 2002); around 1 hPa diurnal variability becomes prominent and local time sampling must be taken into account (Sakazaki et al., 2015). In

contrast, water vapor retrievals from the various satellite instruments exhibit biases of 20% or more relative to one another, depending on the combination of instruments (Kley et al., 2000). Thus, a key aspect of any long-term WV data merging is the requirement for some homogenization procedure to account for measurement offsets. In SWOOSH, this is accomplished by calculating instrument offsets using satellite-satellite coincident measurements taken during overlap time periods.

The rest of this paper is organized as follows: In Sect. 2, we present the satellite data sets used in this study and discuss the screening that has been applied to the data. In Sect. 3, we discuss the process of choosing a reference





satellite instrument to which other data are adjusted. In Sect 4., we describe the data homogenization, gridding, and combining process, including uncertainty estimation. Finally, in Sect 5, we present examples of the utility of the long-term record for capturing seasonal to interannual variability.

**2 Data and basic screening**

In this section, the SWOOSH satellite input data and screening procedures are described. Currently, SWOOSH contains data from several satellite instruments: SAGE II, SAGE III, HALOE, UARS MLS, and EOS Aura MLS (hereafter Aura MLS). Basic information about these instruments including their operating periods and vertical resolutions is shown in Fig. 1 and Table 1. This subset of available satellite data was chosen because each instrument provides vertical profiles of both $O_3$ and WV with roughly similar vertical resolution, and

because this combination of instruments provides overlapping coverage from 1984 onwards. Although there are several other satellite instruments that could be added during the 2000's, these data are not included because the Aura MLS provides sufficient sampling and global coverage on a monthly timescale.

Below, a brief description of each satellite instrument and the "basic" data screening is given. Basic screening is based on the published recommendations of the satellite instrument teams. Additional screening criteria based

on satellite-sounding matched data comparisons are given in Appendix A.

**2.1 SAGE II and SAGE III**

The Stratospheric Aerosol and Gas Experiment (SAGE) instruments provide water vapor, ozone, and aerosol profiles from measurements of solar radiation attenuated through the Earth's limb during sunrise/sunset events viewed from the satellites' orbits. SAGE II was launched in October 1984 aboard the Earth Radiation Budget

Satellite (ERBS), and made measurements spanning 80°S – 80°N until August 2005. SAGE II was a seven-channel photometer measuring in the range from 0.385 μm to 1.02 μm (McCormick, 1987), with water vapor (ozone) retrieved from the channel at 0.935 μm (0.6 μm). SAGE III was launched into a Sun-synchronous orbit (100° inclination) on December 10, 2001, aboard the Russian Meteor 3M platform. In this orbit, solar occultation measurements were made at 47-84° latitude in the NH, and 31-57° in the SH. SAGE III operated

until November 2005, measuring in 87 channels ranging from 0.290 to 1.54 μm.

Basic information about the SAGE and other satellite instruments, including the vertical resolution and range for the ozone and water vapor products, is listed in Table 1. Here, we use version 7.0 data from SAGE II (Damadeo et al., 2013), which is an update from the prior version (version 6.2, Thomason et al., 2004). Version





7.0 includes updated ozone spectroscopy to be consistent with the most recent SAGE III processing, and a new determination of the water vapor channel filter parameters. We exclude SAGE II water vapor data before January 1, 1986 due to a known drift in the water vapor channel filter parameters during this time period (Thomason et al., 2004). For SAGE III, we use version 4 data (Thomason et al., 2010).

The SAGE II data are retrieved on a 0.5 km grid from the surface to 50 km for WV and 0.5 to 70 km for $O_3$. The SAGE sampling corresponds to a Nyquist limited vertical resolution in transmission of 1 km. Ozone is unsmoothed and thus retains a 1 km resolution. Water vapor, however, is smoothed variably in altitude and so the actual vertical resolution ranges from 1 to 3 km (Damadeo et al., 2013). For SAGE III both WV and $O_3$ are retrieved on a 0.5 km grid from 0 to 100 km. The SAGE III water vapor resolution is 1.5 km (Thomason et al.,

2010), and the ozone resolution is 1 km (Hassler et al., 2014).

SAGE II water vapor data are filtered according to the published recommendations of Taha et al. (2004) and Rind et al. (Rind et al., 2005) to remove poor quality retrievals and profiles impacted by high volcanic aerosol loading (e.g., following the eruption of Mt. Pinatubo in 1991). Specifically, we remove any points with water vapor uncertainty > 50%. Additionally, we remove all data in a profile below the highest altitude point at which

either cloud presence is flagged or $\beta_{1020}$ (1020 nm extinction) > 2 x $10^{-4}$ km$^{-1}$ and all profiles associated with "short events" during 1993-1994, as described in Taha et al. (2004).

Because the above screening fails to remove all data with clearly unphysical values, we apply additional outlier screening. First, we remove extreme outliers, defined as $H_2O$ mixing ratio > 30 ppmv above 100 hPa. Then, we remove points farther than $3\sigma$ from the mean in 10° latitude bins. This results in the removal of an additional

0.6% of the $H_2O$ data, with 80% of the screened data being high outliers.

For SAGE II ozone, we filter data based on the recommendations of Wang et al. (2002). As with the SAGE II water vapor, this filtering removes aerosol contaminated and poor quality retrievals. The filtering involves discarding all points in a profile below which any of the criteria are met: $\beta_{525}$ (525 nm extinction) > 6 x $10^{-3}$ km$^{-1}$, or 1 x $10^{-3}$ < $\beta_{525}$ < 6 x $10^{-3}$ km$^{-1}$ and $\beta_{525}/\beta_{1020}$ > 1.4, or cloud presence flagged, or uncertainty equals 300%.

Additionally, any profile that contains an uncertainty > 10% between 30 and 50 km is removed. Finally, the same $3\sigma$ filtering is applied to ozone as with the water data. This results in an additional removal of 0.6% of the ozone data, with 70% of the screened data being high outliers.

For SAGE III data, the data were pre-screened by the retrieval team, so with one exception no additional screening is necessary. The only screening applied to SAGE III data is removal of a few weeks worth of bad

data during 2002, following the recommendations of Thomason et al. (2010). We also apply a $3\sigma$ filtering, as wit the SAGE II data. This results in removal of 0.5% of the water vapor data and 0.3% of the ozone data.





### 2.2 UARS MLS

The Upper Atmospheric Research Satellite (UARS) was launched into a near circular (57°) orbit in September 1991, and operated until November 2005. The UARS Microwave Limb Sounder instrument measured emitted microwave radiation from the Earth's limb using three heterodyne radiometers centered at 63, 183, and 205 GHz (Barath et al., 1993). Latitudinal coverage of UARS MLS was either 34°S – 80°N or 34°N – 80°S, depending on the spacecraft yaw position which switched every 36 days.

Water vapor profiles in the stratosphere and mesosphere were retrieved using the 183 GHz radiometer, which operated from September 1991 until April 1993. We use the UARS MLS version 6 data (Pumphrey, 1999; Pumphrey et al., 2000), which is retrieved at 6 levels per decade of pressure (resulting in ~2.5 km vertical resolution) and produces useable profiles between about 100 – 0.1 hPa. The UARS MLS vertical resolution is approximately ~3-4 km throughout most of the stratosphere (Pumphrey, 1999). Profiles containing negative uncertainty values are removed, which is equivalent to filtering out any of the bad MMAF_STAT quality flags related to unstable or unphysical retrievals (http://browse.ceda.ac.uk/browse/badc/mlsl3/data/NERC-Edinburgh-h2o/v0006-edinburgh-h2o-official/00README_for_v0104).

Ozone was retrieved separately from both the 183 GHz and 205 GHz channels, but we use only the 205 GHz version 5 ozone; it is the recommended product for stratospheric ozone, and it extends through the end of the UARS mission, unlike the 183 GHz product (Livesey et al., 2003). Like the water vapor product, ozone is retrieved at 6 levels per decade of pressure. Ozone data are filtered based on the criteria of Livesey et al. (2003). In particular, only data with QUALITY_O3_205=4 are used, and data with negative uncertainty or bad MMAF_STAT quality flags are removed. Also, only ozone data between 100 hPa and 1 hPa are used.

### 2.3 UARS HALOE

In addition to UARS MLS, the Halogen Occultation Experiment (HALOE) operated on board UARS from October 1991 until November 2005, making solar occultation measurements in the infrared (2.45 – 10.0 μm) with latitudinal coverage from 80°S to 80°N (Russell III et al., 1993). HALOE contains 6 broadband channels for simultaneous retrieval of water vapor (6.61 μm) and ozone profiles (9.85 μm) from approximately 10 – 90 km. HALOE data are retrieved on a ~0.5 km grid from 1000 hPa to $1 \times 10^{-6}$ hPa (271 levels). The HALOE vertical resolution is 2.3 km for water vapor and ozone (Harries et al., 1996; Kley et al., 2000). Here, we use the HALOE version 19 (v19) products, which have been extensively compared with independent satellite, balloon, and ground-based data (Kley et al., 2000; Nedoluha et al., 2007).





The HALOE ozone and water vapor data are filtered by first removing any "trip angle" or "constant lockdown angle" events identified by the data providers (http://haloe.gats-inc.com/user_docs/index.php). Then, any points with uncertainties ≥100% are removed (E. Remsberg, personal communication). Finally, we apply an additional aerosol screening procedure to remove affected profiles after the eruption of Mt. Pinatubo. Briefly, we remove

any profiles where the NO-channel extinctions at 15 hPa is greater than $4 \times 10^{-5}$ km$^{-1}$. The reasoning for this is illustrated in Fig. 2, which shows the plot of "total water" ($H_2O + 1.6*CH_4$) versus HALOE NO-channel extinction for tropical data (30°S-30°N) at a number of different pressure levels. "Total water" is roughly constant in the tropical stratosphere (e.g., see Figure 7, Letexier et al., 1988), and there are no physical reasons that it should depend on aerosol extinction. Indeed, at lower extinction values, there is no correlation between

total water and extinction. However, at 15 hPa there is a clear dependence of total water on extinction for extinction values above $\sim 4 \times 10^{-5}$ km. The mean total water values at each pressure level are listed in Fig. 2 for profiles with $\beta_{15hPa}$ (NO-channel extinction at 15 hPa) greater than or less than $4 \times 10^{-5}$ km$^{-1}$. From these numbers, it is apparent that profiles with high extinction at 15 hPa are dry biased at the upper levels, and wet biased at the lowest levels in the stratosphere. To remove these events, we discard the profile below 15 hPa

when $\beta_{15hPa} > 4 \times 10^{-5}$ km$^{-1}$. All profiles screened by this new algorithm occurred before November 1992 in the first year of operation of HALOE, when volcanic aerosols from the Mt. Pinatubo eruption heavily affected the profiles.

### 2.4 EOS Aura MLS

The Aura MLS instrument was launched in July 2004 aboard NASA's Earth Observing System (EOS) Aura

satellite (Waters et al., 2006). The Aura MLS instrument measures thermally emitted microwave radiation from the Earth's limb using five radiometer channels spanning 118 GHz to 2.5 THz; ozone is retrieved from the 240 GHz channel (Froidevaux et al., 2008; Jiang et al., 2007b; Livesey et al., 2008), and water vapor is retrieved from the 190 GHz channel (Lambert et al., 2007; Read et al., 2007). The Aura MLS obtains ~3500 profiles per day and achieves nearly global coverage between 82°S and 82°N. Vertical resolution generally increases as a

function of height in the stratosphere for both retrievals, but is 2.8-3.5 km for water and 2.5-3 km for ozone between 100 hPa – 1 hPa. Over this same range, the estimated accuracy (100 – 1 hPa) for ozone and water vapor is ~5-10%. For both products, we use version 4.2 data, which is provided at 12 levels per decade (~ 1.25 km) below 1 hPa.

The data filtering for the Aura MLS data is similar to that for version 2.2 data discussed in Lambert et al. (2007)

for water vapor and Froidevaux et al. (2008) for ozone, but with updated values presented in the Aura MLS





version 4.2 data quality document (http://mls.jpl.nasa.gov/data/v4-2_data_quality_document.pdf). Additional filtering of the Aura MLS WV data in the UTLS is described in Appendix A. This filtering is motivated by the dry bias present in the MLS data, as described in the next section.

### 3 In situ balloon measurements vs. satellite observations: Choosing a reference satellite measurement

In this section, we compare balloon-borne ozone and frostpoint hygrometer sounding measurements to coincident satellite observations. These comparisons are used to quantify biases of the various satellite measurements, to identify additional filtering that is needed for the satellite data, and to justify our choice of Aura MLS for water vapor and SAGE-II for ozone as the reference measurements to which other measurements are adjusted.

#### 3.1 Frostpoint hygrometer comparisons

The frostpoint measurements analyzed here are comprised of 1438 NOAA Frostpoint Hygrometer (FPH, Hurst et al., 2011) and Cryogenic Frostpoint Hygrometer (CFH, Vömel et al., 2007b) balloon soundings taken as part of both routine monitoring and field campaign work between the beginning of the SAGE-II measurements in October 1984 and January 2015 (station details given in Table 2). Data include routine monitoring from Boulder, CO; Hilo, HI; Lauder, NZ; and Lindenberg, Germany. The field campaign projects include the Soundings of Ozone and Water in the Equatorial Region (SOWER) project at sites in Indonesia, Vietnam, Kiribati, and Ecuador (Hasebe et al., 2013; Fujiwara et al., 2010), the Ticosonde project at San Jose in Costa Rica (Fujiwara et al., 2010; Selkirk et al., 2010), the Sounding Water vapor, Ozone and Particle (SWOP) project at sites in China except for Yangjiang (Bian et al., 2012), the 8th World Meteorological Organization (WMO) Intercomparison of High Quality Radiosonde Systems at Yanjiang, China (Nash et al., 2011), and the research vessel Mirai campaign in the tropical Indian Ocean (Suzuki et al., 2013).

Matched satellite profiles are found by searching for any profiles within a given time, distance, and equivalent latitude range of the frostpoint (FP) data. Potential vorticity (PV)-based equivalent latitude from the MERRA reanalysis (Rienecker et al., 2011) is used as a match criterion because it allows for similar air masses to be identified (e.g., Manney et al., 2007). For all data except Aura MLS, the match criteria used are ± 2 days, ± 2000 km E-W distance (± 18° longitude at the Equator), and ± 1000 km N-S distance (± 9° latitude). Due to the significantly better horizontal coverage for Aura MLS, the match criteria are stricter: ± 0.75 days, ± 1000 km E-



W, and ± 500 km N-S, as used in Hurst et al. (2014). Additionally, for all datasets, we require the average equivalent latitude (between 316 hPa and 68 hPa) to be within 5° of the corresponding average equivalent latitude from the FP. If more than one profile meets the above criteria, the profile with the closest equivalent latitude to the FP measurement is chosen. Using these match criteria, 1150 of the FP profiles are identified as being matched with one or more satellite measurements.

For direct comparison with the satellite data, the FP data are averaged from their native resolution to that of the satellite using either the averaging/smoothing operators (for Aura MLS) or a triangular smoothing with Full Width at Half Maximum (FWHM) equal to the instrument resolution (see Table 1), and then all datasets are interpolated onto the 12 levels per decade Aura MLS vertical grid for comparison purposes. Finally, for each level, any outlier pairs that are more than 5 standard deviations from the mean percent difference between the satellite and FP data are removed.

At and above 100 hPa, agreement between the satellite and FP WV data is generally quite good (Fig. 3): it is within 10% except for HALOE and UARS MLS data, which are ~15-25% dry biased relative to the frostpoint hygrometers in the lower stratosphere. At lower levels (higher pressure), the satellite biases generally become larger and drier relative to the FP data.

As can be seen in Fig. 3, HALOE data are extremely dry biased (values < 50% of the FP data) below 147 hPa. This extreme dry bias occurs because HALOE retrieval "saturates" under tropospheric conditions when the WV values are above ~10 ppmv. The 147 hPa level is a transition region in water vapor where the values can range from tropospheric to stratospheric, depending on season and latitude. This is reflected in the HALOE/FP comparison at 147 hPa (not shown): HALOE data show mean dry biases ~50% when FP values are greater than 8 ppmv, as opposed to a ~10% dry bias for values less than 8 ppmv. To avoid a (dry-)biased monthly mean from HALOE, we exclude HALOE data at and below the 147 hPa level.

The agreement between the SAGE instruments and the FP data is not as well constrained due to the small number of matches (N ~ 20, with even fewer matches at the lowest levels), but SAGE-II data exhibit a ~20-50% dry bias below 200 hPa, possibly due to $O_3$ interference (Damadeo et al., 2013). SAGE-III also shows a dry bias relative to FP data, but is within 20% of the FP above 215 hPa.

At and above 121 hPa, Aura MLS data are within 7% of the FP values, in broad agreement with previous findings (Lambert et al., 2007; Read et al., 2007; Hurst et al., 2014; Vömel et al., 2007a). At lower levels, the Aura MLS data exhibit a dry bias that varies with pressure and peaks at 36% at 215 hPa. Our result at this level is similar to estimated 25% bias given in Read et al. (2007) and 27% given by Vömel et al. (2007a), and is further explored in Appendix A.





At all levels the Aura MLS and FP measurements are well within 2 standard deviations (2σ) of their mean difference at the given level. A more rigorous metric, used in Fig. 3, is the standard error of the mean differences ($\sigma/\sqrt{N}$). For the null hypothesis that the measurement (i.e., population) means are equal and no systematic uncertainty exists in either measurement, the mean difference between the measurements (i.e., the

sample mean difference, $\bar{X}$) is t-distributed. Then, the null hypothesis can be rejected if $\bar{X} > t_{crit}\dfrac{\sigma}{\sqrt{N}}$, where

$t_{crit} \approx 2$ (two-tailed, p = 0.05). Hence, in Fig. 3 levels at which the $2\sigma/\sqrt{N}$ shading intercepts zero can be said to be in agreement with the FP data.

It is worth noting that in Fig. 3 we define the percent difference of the satellite data relative to the mean value of the satellite and FP data (i.e., percent difference = (sat-FP)/((sat+FP)/2)*100). If the percent difference is

defined relative to the frostpoint only (i.e., (sat-FP)/FP*100), there is an asymmetry whereby the percent difference is constrained to be ≥ -100%, but is unconstrained in the positive direction. Computing the percent change relative to one of the instruments causes the distribution of percent difference at each level to be skewed toward positive values and produces positively biased estimates of the mean.

The difference in results from these two definitions is greatest below 100 hPa, where there is inherently greater

variability in WV and a greater likelihood of mismatched profiles. In that region, it is much more likely that one of the pair of "matched" profiles is of dry stratospheric air and the other is of wet tropospheric air. Also, all satellites considered here have a large horizontal footprint (~ hundreds of kilometers) compared to the frostpoint measurements, so this could add to the observed differences. In this region, the mean and median percent differences are very different from one another (often over ~50%) for the conventional definition (i.e., (SAT-

FP)/FP*100), but are almost the same using the definition implemented in Fig. 3.

It is clear that at many levels neither the Aura MLS nor the other satellite instruments are in agreement with the FP data, as defined by the $2\sigma/\sqrt{N}$ criteria. However, there are several important considerations in interpreting this disagreement. First, imperfect matching (in time and space) between the measurements likely contributes to a difference. In the UTLS spatial and temporal matches may not be as good because the intrinsic variability of

WV mixing ratios is high and spans several orders of magnitude. Also, the statistical test does not account for seven though the aforementioned statistical method is valid for quantifying the significance of mean biases in the satellite measurements, for some tasks a more relevant question is which satellite provides the least biased measurement over a broad range of conditions. This question is inherently subjective, but for the purposes of



constructing a monthly-mean gridded WV database the Aura MLS is chosen as a reference instrument because the dataset has been extensively validated and contains a relatively small mean bias over a wide range of pressure levels. However, it is clear that there is a significant Aura MLS UTLS dry bias at and below 147 hPa. Additional screening of the Aura MLS data set in the UTLS is discussed in Appendix A.

**3.2 Ozonesonde comparisons**

Here, we use ozonesonde data from a subset of 11 stations with high quality measurements spanning a broad range of latitudes. These 11 stations are listed in Table 3, and their data were obtained from the World Ozone and Ultraviolet Radiation Data Centre (WOUDC) and Southern Hemisphere ADditional OZonesondes (SHADOZ) project (Thompson et al., 2012). The criteria chosen for these stations are that all stations use the

Electrochemical Concentration Cell (ECC) type ozonesondes and provide a data record that extends back to the start of the SAGE-II record in 1984. At some stations ozonesonde types have been changed over the years, but only ozone profiles obtained with ECC sondes were used for the comparison here, as they have a documented accuracy of 5-10% below 30 km (Smit et al., 2007). Additionally, the sounding stations were selected to cover the Polar Regions and the mid-latitudes of both Hemispheres, as well as the tropics (see Table 3).

The temporal/spatial match criteria, outlier screening, and vertical smoothing used is the same as for the frostpoint hygrometer comparison. In addition, we manually filter out obvious outlier sonde profiles with unphysical values that were not properly quality controlled, and apply an additional outlier-screening algorithm that consists of removing ozone concentrations falling $2\sigma$ outside of the mean value for each month and each station on a 0.25 km vertical grid.

The resulting vertical profile of differences between the satellite data and ozonesondes is shown in Fig. 4. In general, the agreement between satellite and ozonesonde data is much better than for the corresponding water vapor comparison, with all instruments falling within ±20% of the ozonesonde data above 100 hPa. Overall, SAGE-II shows the best comparison with the ozonesonde data over the range of the stratosphere, and most of the instruments diverge from the ozonesonde data in the UTLS region around 200 hPa. For this reason, we use

SAGE-II as our reference ozone measurement to which other ozone measurements are adjusted.

**4 Dataset construction**

In this section, we describe the methodology for homogenizing, gridding, and merging the satellite data to create the combined SWOOSH data product. Briefly, the homogenization process involves adjusting data from the





individual satellite instruments to create statistical agreement with the reference satellite (i.e., SAGE-II for ozone and Aura MLS for WV). After the homogenization process, data from each instrument is gridded individually, and the individual fields are merged into the combined product.

**4.1 Removing instrument bias with instrument offsets**

In general, satellite measurements of the same quantity such as WV may not agree with one another, even when the measurements are close in time and space and are nominally in the same air mass. In this hypothetical scenario, the difference between a single matched pair of satellite measurements can be explained as a combination of "bias" and "noise". Bias could be caused by systematic errors in the underlying retrieval(s), or due to spatial resolution differences of the measurements (especially in the vertical). On the other hand, random

measurement uncertainty (precision), instrument spatial resolution differences (e.g., in the horizontal), and imperfect matching in space/time between the two measurements can be thought of "noise". The key distinction between "bias" and "noise" in this context is that when averaged over a sufficiently large number of matched pairs of measurements, the average inter-satellite differences due to "noise" tend towards zero (by definition), while any statistical bias between the measurements does not diminish because it is related to a fundamental

difference in the measurements.

In SWOOSH, pairs of matched measurements between the reference satellite and the non-reference satellite measurements are used to calculate the mean offset of the non-reference data for its full measurement period. This instrument offset is then added to the non-reference satellite data to achieve statistical agreement with the reference data.

The matching methodology for the inter-satellite matches is the same as in Sect. 3 for the comparison between balloon sounding data and satellite data. Because of the relatively sparse sampling of the solar occultation measurements, we use the less strict criteria described in Sect. 3: specifically, we use the pair of measurements with closest equivalent latitude that is within ± 2 days, ± 2000 km E-W distance (± 18° longitude at the Equator), and ± 1000 km N-S distance (± 9° latitude) from one another. With these criteria, the number of matched pairs

ranges from ~5,000 – 25,000 depending on the specific combination of instruments.

The matching between the reference and non-reference data set for each species is possible for all combinations of data sets except between Aura MLS and UARS MLS for water vapor, since their records don't overlap. In the absence of instrument overlap for these two instruments, we use the (adjusted) HALOE WV as a transfer standard.



After matching, we interpolate all satellite data onto the SWOOSH vertical grid, which corresponds to the Aura MLS vertical grid containing 12 levels per decade. This grid is coarser than the retrieval vertical grids of all of the other instruments except for UARS MLS, which has roughly half the number of vertical levels as Aura MLS. It is worth noting that the vertical resolutions of the satellite instruments are not the same, and range from about

$1 - 5$ km depending on the species, instrument, and vertical level considered (see Table 1). We conducted tests smoothing the higher resolution data down to the ~3 km resolution of Aura MLS, and did not find large changes in the computed offsets, even near the tropopause, indicating that the offsets are caused more by fundamental retrieval issues than simple differences in vertical resolution of the measurements. For simplicity, we use linear interpolation in log-pressure space to put all satellite data sets on the Aura MLS levels. Satellite data sets on a

native altitude-number density coordinate system (i.e., SAGE II and SAGE III) have all been converted to pressure-mixing ratio coordinates using temperatures from the MERRA reanalysis.

After matching and interpolating to a common vertical grid, the mean offset is calculated for each vertical level and 10° latitude bin for each combination of satellite instruments. Figures 5 and 6 show the offsets as a function of height and latitude for water vapor and ozone, respectively. It is worth stressing that the offsets added to the

non-reference satellite data here do not vary with time or season, only with height and latitude. Thus, drifts in individual satellite records, if they exist, are not accounted for in SWOOSH.

In addition to the instrument offsets, we also compute the uncertainty in the offsets. Since the offset is defined as the mean difference between coincident pairs of satellite instruments within a 10° latitude bin at a given pressure level, the offset uncertainty is simply the standard error of this mean difference (i.e., $\sigma / \sqrt{N}$ ). This is

illustrated in Fig. 7, which shows the 68 hPa level offset versus latitude for HALOE/Aura MLS, and the histogram of mean difference between the two at one latitude band.

To further illustrate the impact of the offset adjustment process and its associated uncertainties, Figures 8 and 9 show example time series of water vapor and ozone, respectively. These figures show the time series before and after the homogenization process to illustrate the magnitude offsets and their uncertainty.

**4.2 Gridding**

SWOOSH is produced on several different horizontal and vertical grids to serve different user needs. For a given horizontal/vertical SWOOSH grid (summarized in Table 4), the data from all species and satellites are stored in a single file with a monthly time resolution. On each horizontal grid for each species/satellite/month, SWOOSH contains several different monthly statistics, including the mean mixing ratio for both the "raw" and





"adjusted" versions of the data, the number of profiles, their standard deviation, and a measure of the combined retrieval (precision) and offset uncertainties.

The uncertainties stored in SWOOSH for each species are the root-mean-sum-of-squares combination of the retrieval precision uncertainty and offset adjustment uncertainty. A full description of the SWOOSH source
record uncertainty is provided in Appendix B.

The primary SWOOSH grid is a zonal-mean gridded data set (either 2.5°, 5°, or 10° latitude) on pressure levels (12 levels per decade from 316 hPa to 1 hPa, corresponding to the Aura MLS pressure levels). Although data at pressure levels above 1 hPa are available in most of the source data sets used in SWOOSH, the diurnal cycle in ozone becomes prominent above this level. The SWOOSH record makes no attempt to quantify biases that may
be related to diurnal sampling, or to non-uniform spatial or temporal sampling within a monthly latitudinal grid box. Depending on the magnitude of the seasonal or sub-monthly gradients, uneven sampling could introduce additional systematic error beyond what is accounted for in the SWOOSH uncertainty estimates (Damadeo et al., 2014; Neely et al., 2014).

Additional SWOOSH grids include a coarsely gridded 3D (i.e., longitudinally resolved) product on pressure
levels, and a zonal mean product on an isentropic vertical grid. For creating the isentropic grid, we use the closest 6-hourly MERRA reanalysis temperature profile to each satellite measurement to compute potential temperature at the satellite vertical grid, and then interpolate the satellite data on to the theta grid. The output potential temperature grid ranges from 300 to 400 K in 10 K increments, and 400 to 650 K in 25 K increments.

For the zonal mean SWOOSH grids, SWOOSH variables are provided on an equivalent latitude grid (in
addition to the standard geographic latitude grid). Here, equivalent latitude is defined using potential vorticity (PV) on an isentropic ($\theta$) surface, as used in numerous previous studies (e.g., Nash et al., 1996; Butchart and Remsberg, 1986). At a specific longitude $\lambda_o$, latitude $\phi_o$, and isentropic level $\theta_o$, the PV-based equivalent latitude ($\phi_{eq}$) is the latitude at which the area poleward of $\phi_{eq}$ is equal to the area where $PV(\lambda,\phi) \leq PV(\lambda_o,\phi_o)$. Compared to a geographic latitude coordinate system, tracers such as $WV/O_3$ in a PV-based equivalent latitude
coordinate system contain less variability, as North-South excursions in the tracer field are due to largely reversible synoptic-scale features.

The potential vorticity (PV)-based equivalent latitude is computed from the 6-hourly MERRA PV data as above for each satellite profile. Data on the equivalent latitude grid are likely to be most useful in polar process studies. As can be seen in Fig. 10 for ozone, the signature of Antarctic stratospheric ozone depletion is much clearer on
an equivalent latitude grid than on a geographic latitude grid. Also, at many levels improved horizontal coverage is achieved on the equivalent latitude grid relative to the geographic grid. Outside of the vortex, the equivalent





latitude gridded data is very similar to the geographic latitude gridded data, as expected. The similarity between these two grids is exploited in section 4.5 to use the equivalent latitude gridded data to fill in data missing from the geographic grid.

**4.3 The combined monthly-mean product**

After all of the individual satellite data sets have been offset-adjusted and gridded, the combined SWOOSH product is formed from the source data records. For a given month/latitude/level, data from all available satellite instruments are combined using a weighted average based on the number of observations from each instrument, i.e.,

$$\bar{q} = \sum_{k=1}^{K} \bar{q}_k \frac{N_k}{N_{tot}}$$

where $\bar{q}_k$ is the monthly-mean mixing ratio of the $k^{th}$ satellite in the bin, $N_k$ is the number of observations in the bin, K is the number of satellites, and $N_{tot} = \sum_{k=1}^{K} N_k$.

By combining data in this way, the combined product is dominated by the Aura MLS measurements after their introduction in August 2004, as Aura MLS contains more than an order of magnitude more data in a monthly grid box than all of the other datasets combined. In the pre-Aura MLS period the data density is often low for a single instrument in a given 10° latitude band, so combining data using a weighted mean based on the number of available measurements (rather than simply averaging the two monthly means, e.g.) gives a more representative value.

**4.4 Uncertainty in the combined monthly-mean product**

In addition to the combined monthly-means, an uncertainty and standard deviation of the combined product is also provided. The derivation and details of the combined uncertainty and standard deviation estimates are provided in Appendix B. As shown in the appendix, the combined uncertainty ($\sigma_{\bar{q}}$) is the standard error of the combined monthly mean ($\bar{q}$), which can be expressed as a combination of the individual satellite uncertainties ($\sigma_{\bar{q}_k}$). The individual satellite uncertainties include contributions from the satellite retrieval uncertainties (which are provided by the individual satellite teams) and the offset adjustment uncertainties (from Sect. 4.1). Also stored is the standard deviation of the source measurements contributing to the monthly mean ($s_k$) as well as a standard deviation for the combined product ($s$). Figures 8 and 9 illustrate the individual satellite standard





deviations and uncertainties, the combined uncertainty, the combined standard deviation, and the offset uncertainties.

One issue that stands out in these figures is that while the individual satellites have quite similar standard deviations to one another, their retrieval uncertainty estimates vary wildly, particularly for water vapor. For

5 example, the HALOE WV uncertainties are extremely small relative to the other instruments, whereas the SAGE-II WV uncertainty estimates are relatively large. Furthermore, the HALOE (WV) uncertainties are much smaller than the corresponding HALOE standard deviations, and the SAGE-II uncertainties are larger than the SAGE-II standard deviation by a factor of ~3 (see discussion below).

It is worth noting that in general, the "observed" standard deviations ($s_k$) should be of similar magnitude or

10 greater than the instrument precision uncertainty (i.e., the random uncertainty). For example, if geophysical variability (i.e., the population standard deviation for a given height/latitude/month bin) is small relative to the instrument uncertainty, then the observed (sample) standard deviation should be close to the instrument uncertainty. If, on the other hand, geophysical variability is larger than the instrument uncertainty, then the observed standard deviation is a mixture of both instrument precision and geophysical variability.

15 For the case of water vapor at the 68 hPa level illustrated in Fig. 8, the Aura MLS WV monthly precision uncertainties and standard deviations are of the same magnitude. This result is consistent with Lambert et al. (2007), who demonstrated that the "observed" and "expected" (i.e., based on instrument precision) standard deviations of coincident pairs of Aura MLS profiles from ascending and descending orbit matches are in close agreement (e.g., their Fig. 3). In their case, the reason for the similarity is that the ascending/descending orbit

20 matches measure the same air mass twice; the "observed" difference between the two measurements is dominated by random measurement errors associated with instrument imprecision.

In our case, the fact that the Aura MLS WV uncertainties and standard deviations are so similar suggests that geophysical variability (i.e., over a 10° latitude band within a month) is small relative to the Aura MLS uncertainty. This is in contrast to the Aura MLS $O_3$, where the standard deviation is clearly larger than the

25 uncertainty (Fig. 9) because of significant $O_3$ variability.

The mismatch of the WV instrument uncertainties and standard deviation in SAGE-II and HALOE data has important implications for the combined uncertainty estimates in SWOOSH. It is possible that the HALOE uncertainties are underestimated and it is likely that the SAGE-II uncertainties are overestimated, which leads to an artificially inflated or deflated SWOOSH combined WV uncertainty estimate before August 2004, depending

30 on which instrument contributes more data to a given lat/month/height bin. As discussed in Damadeo et al. (2013), the most likely explanation for the overestimated SAGE II uncertainties is the inclusion of additional

aerosol clearing uncertainty, which inflated the water vapor uncertainty. Because of these potential issues, for an uncertainty estimate during the pre-2004 period we recommend using the combined standard deviation ($s$, Eq. B16) instead of the combined uncertainty value ($\sigma_{\bar{q}}$, Eq. B7) stored in SWOOSH.

### 4.5 Additional SWOOSH products: climatology, anomaly, and filled data

Depending on the scientific objectives, it is desirable, or indeed required in some cases, to have a dataset that is free of missing data. In other cases, the focus may be the climatological seasonal cycle, or departures from the seasonal cycle (anomalies). SWOOSH includes several data products to fulfill these needs.

For most variables including the individual satellite data and the combined product, there is both a seasonal cycle and an anomaly time series provided. The anomaly time series simply has the long-term mean seasonal

cycle (computed over the entire record) removed at each time/grid box.

There are three categories of filled data products in SWOOSH: "equivalent latitude filled", "anomaly filled", and "equivalent latitude filled" + "anomaly filled". The "equivalent latitude filled" products are simply the geographically gridded variables with missing data filled in using data from the corresponding equivalent latitude gridded data (i.e., in the same latitude/height/month bin). This is useful for filling in data near the poles.

As an example, Aura MLS only samples ±82° latitude, so any grid boxes poleward of these latitudes would contain no data in the geographically gridded version of the variables. However, in any given month, Aura MLS typically samples air masses with an equivalent latitude poleward of 82°, as reflected in the equivalent-latitude gridded version of the data in Fig. 10. It is worth noting that in the example of polar ozone loss this filling is likely to add values in that are an underestimate relative to the true geographical zonal means, because inside the

polar vortices the ozone at a given equivalent latitude is less than the corresponding geographic latitude. However, at other latitudes and for water vapor where the geographic and equivalent latitude latitudinal distributions are similar, the procedure does not introduce a significant bias. Other than at the most poleward gridpoints, the "equivalent latitude filled" combined product is only significantly different than the regular combined product in the pre-Aura MLS period when there are large (latitudinal) gaps in the data in any given

month.

In addition to the equivalent latitude filled version of the data, which in general is not a gap-free data set, SWOOSH also includes an "anomaly filled" version of the data (see Fig. 11) that is free of missing data. In this version, the anomaly data (Fig. 11b) are first filled in the latitude-time plane (separately at each vertical level) using radial basis function interpolation with an inverse multiquadric function (Fig. 11c). In such an

interpolation, the interpolated value is based on a mean of nearby points weighted by the inverse of their





distance. Because the poles contain missing data throughout the record, we fix the anomaly values at zero at the poles. Doing this ensures that the "anomaly filled" field will relax towards zero from the most poleward valid value in the anomaly field. After this step, the anomaly array (Fig. 11c) is simply added back to the corresponding seasonal cycle array to produce what is known as the combined "anomaly filled" version of the SWOOSH data (Fig. 11d).

As a final note, we stress that the anomaly filled version of the SWOOSH data represents only one way of creating a filled dataset, and that where trend analysis is of interest the unfilled version of SWOOSH should be used for analysis, partly out of caution and partly because it contains reliable uncertainty estimates that can be used for trend uncertainty estimation. The anomaly filled data should in general be used with extreme caution in the pre-1990 time period. Regions with very sparse and noisy data can have undue influence over large regions in the filling process, as can be seen in the high southern latitudes in the example shown in Fig 10b-c.

## 5 Examples of variability

In this section, we demonstrate the utility of SWOOSH for quantifying and studying seasonal and interannual variability in stratospheric water vapor and ozone. SWOOSH is used to illustrate several well-known ozone and water vapor phenomena such as the tropical tape recorder, transport of ozone and WV anomalies in the lowermost stratosphere, and variability related to the QBO.

### 5.1 Tropical tape recorder and ozone QBO related variability

The tropical tape recorder signal (Mote et al., 1996) is one of the fundamental manifestations of the seasonal to interannual variability in the stratospheric water vapor distribution. Figure 12 shows the tropical tape recorder signal from the individual satellite instruments, as well as the combined anomaly-filled SWOOSH product, and the tropical tape recorder anomaly. As can be seen from this plot, the combined data clearly captures the post-2000 drop in WV, as well as significant interannual variability. Using the combined product, we compute the post-2000 drop in water vapor to be 0.4 ppmv (averaged 30°S – 30°N, (2001.5-2005.5) minus (1996-2000)), similar to the values found in other studies (e.g., Randel et al., 2006; Solomon et al., 2010).

Similar to the tape recorder plot, Fig. 13 shows the deep tropics (10°S – 10°N) averaged ozone anomalies as a function of height and time. The descent of ozone anomalies associated with the QBO can be clearly seen in this figure. Numerous studies have identified QBO-related variations in ozone concentration and column amount





(e.g., Angell and Korshover, 1964, and references therein; Oltmans and London, 1982; Zawodny and McCormick, 1991; Randel and Wu, 1996).

**5.2 Interannual anomalies in transport of ozone and water vapor**

In this section, we illustrate the utility of SWOOSH for capturing interannual anomalies in $WV/O_3$ that are related to transport in the lowermost stratosphere. Fig. 14 shows the latitude vs. time cross-sections of $WV/O_3$ anomalies at 82 hPa in the lower stratosphere. By removing the seasonal cycle, the poleward transport of tropical WV anomalies is easily seen (Fig. 14a), as are interannual variations (e.g., related to the QBO or ENSO, such as the large El Niño event at the end of 2015). For ozone, the interannual variations in anomalies related to interannual variations in polar ozone loss can be seen (Fig. 14b). For example, the weak Antarctic ozone depletion in 2012 (Klekociuk et al., 2014) can easily be seen, as can the severe Arctic polar ozone loss in 1993 (Larsen et al., 1994), 1995 (Manney et al., 1996), and 2011 (Manney et al., 2011).

**6 Data availability**

Version 2.5 of the SWOOSH data are currently archived for long term storage at the National Centers for Environmental Information (NCEI) at https://data.noaa.gov/dataset/stratospheric-water-and-ozone-satellite-homogenized-swoosh-data-set, with a provisional DOI number (doi:10.7289/V5TD9VBX). Current data are hosted at the NOAA Earth System Research Laboratory (ESRL) SWOOSH webpage at http://www.esrl.noaa.gov/csd/swoosh/.

**7 Discussion**

For understanding interannual to decadal variability in the radiatively important trace species of water vapor and ozone on a global scale, it is necessary to combine data records from satellite measurements made with different measurement techniques, data densities, and resolutions. In this paper, we have documented the construction of a new vertically resolved data record of ozone and water vapor from limb measuring satellites. The SWOOSH method of combining satellite data records involves adjusting the non-reference satellite data sets relative to a reference satellite record through the application of offsets. The offsets that are applied to the non-reference data are allowed to vary as a function of latitude and height, but not temporally, to allow for the possibility that inter-satellite biases vary spatially, and are based on coincident pairs of vertical profiles taken during instrument





overlap time periods. The choice of a reference satellite data set is justified based on the best agreement with independent balloon-based sounding measurements so that the SWOOSH combined values will agree with the balloon measurements in an average sense. The adjustment method used in SWOOSH is conceptually similar to previously used methods for combining satellite data sets (Froidevaux et al., 2015; Randel and Wu, 2007)

except that merging takes place in absolute value space rather than in anomaly space, for the reasons noted above.

It must be stressed that no attempt is made to correct for potential satellite drifts in SWOOSH. In principle it is possible that satellite drift could be accounted for by correcting the individual source record or by applying a time varying offset adjustment to the data. However, currently the ability to assess and construct time-varying

corrections for these data is limited due to the sparse sampling of the solar occultation satellites used in the pre-2004 period and the limited spatial and temporal availability of high quality in situ measurements for comparison (Hurst et al., submitted; Hubert et al., 2015).

After adjustment to the reference measurement, the satellite data are geographically gridded, and a number of important statistics including the monthly mean and variance, and uncertainty estimates are provided for both

the individual source records and the combined product. The individual source records can be analyzed independently, and the necessary data is saved to investigate alternative combinations of the data sets (e.g., excluding one of the satellites).

The SWOOSH record constitutes a unique tool for studying interannual to decadal scale variability in water vapor and ozone. The documentation of data provenance, filtering, and merging methodology presented here

provides a traceable basis for future intercomparison studies addressing the agreement of satellite data with balloon and/or ground-based measurement systems, and will be useful for sensitivity studies addressing the impact of satellite homogenization methodologies. The SWOOSH record may prove useful as input to global models lacking interactive ozone chemistry, and will likely be useful for future studies to quantify the radiative impact of water vapor and ozone variability in the UTLS region.

The SWOOSH data (version 2.5) used in this paper are publicly available through the end of 2015 through the NOAA data catalog at https://data.noaa.gov/dataset/stratospheric-water-and-ozone-satellite-homogenized-swoosh-data-set. The SWOOSH data will continue to be updated as long as new data is available from the Aura MLS instrument or a suitable replacement.

In the future, a new source of water vapor and ozone data will be needed in order to continue the record after the

demise of the Aura MLS instrument. For ozone, the NASA Ozone Mapping and Profiling Sensor (OMPS) on Suomi National Polar-orbiting Partnership (S-NPP) satellite has shown promise as a high quality ozone data set



(Kramarova et al., 2014), and will likely be used to continue the SWOOSH record. This measurement will continue with the launch of Joint Polar Satellite System (JPSS)-2.

In contrast to ozone, currently Aura MLS is the only vertically resolved stratospheric water vapor data set available for input in the SWOOSH record. In the near future, there are plans to deploy a SAGE III instrument on the International Space Station that will be capable of providing vertically resolved water vapor, but with severely reduced sampling compared to Aura MLS. Given the water vapor offsets between the satellite instruments demonstrated here, the potential data gap in the water vapor record would severely impact our confidence in characterizing decadal variability and trends in water vapor. As discussed by Müller et al. (2016) and demonstrated in this paper, it is possible that a global network of balloon borne hygrometer measurements could help serve as a transfer standard between satellites and minimize the impact of a potential water vapor data gap in the satellite record.



**Appendix A: Aura MLS UTLS WV screening**

As shown in Fig. 3, Aura MLS is dry biased in the UTLS at levels below 121 hPa. In this section we will demonstrate that the bias varies with both latitude and season, identify the common manifestations of the bias as oscillations and "spikes" that occur at specific levels, and demonstrate a new algorithm removing affected data.

Figure A1a shows a typical comparison between FP data and matched Aura MLS profiles at high latitude. The FP data in this case are from a CFH sonde launched from Sodankyla, Finland (67°N) on 2008-03-07. The 15 closest Aura MLS profiles that meet the match criteria described in Sect. 3.1 are shown in the figure. The upper two plots in Fig. A1 illustrate that the Aura MLS measurements undergo an oscillation about the FP profile in the UTLS region between ~316 – 100 hPa, with local minima at 215 hPa and local maxima at 147 or 121 hPa.

The oscillation is apparent in the Aura MLS apriori profiles (Fig. A1c). These apriori profiles are a result of three separate retrievals that are constrained to be piecewise continuous at 147 hPa and 316 hPa (Read et al., 2007). The existence of the oscillation in both the apriori profiles and the retrieved profiles suggests that the oscillation is not simply an artifact caused by the Aura MLS vertical averaging kernel being applied to a region of high vertical WV gradients, but is introduced at the apriori retrieval stage. To confirm this, we used the Aura

MLS averaging kernel to the degrade the high-resolution CFH data to the Aura MLS levels using the method described in Read et al. (2007), and found that this process does not introduce an oscillation to the CFH data (Fig. A1d).

Since the procedure for applying the Aura MLS averaging kernel to an FP profile requires as input an apriori profile, we also tested the sensitivity of our results to the use of different apriori profiles by using both the Aura

MLS aprioris and the CFH profile as the apriori input. Even when the Aura MLS apriori profiles containing a UTLS oscillation are used as the apriori input to the smoothing procedure, the output CFH profiles do not contain a large oscillation. This is not surprising given that the integrated averaging kernels at these levels are near unity, implying that retrieved WV values at these levels come from the Aura MLS measurements and not the apriori.

To establish that the Aura MLS dry bias is a significant feature from a monthly-mean and climatological perspective, Fig. A2 shows a zonal-mean cross-section of water vapor from the Aura MLS, Aura High Resolution Dynamics Limb Sounder (HIRDLS version 7, Khosravi et al., 2009), UARS HALOE, SAGE-II, SAGE-III, and Atmospheric Chemistry Experiment Fourier Transform Spectrometer (ACE-FTS version 3.5, Bernath et al., 2005) satellite data for the month of March. The dry bias at 215 hPa identified above is obvious

in the zonal-mean monthly-mean Aura MLS data at high latitudes. Aura MLS data show a pronounced



minimum near 215 hPa that is not seen in the other measurements. The positive peak of the oscillation at 147 hPa is not obvious in Fig. A2, in part because the peak sometimes occurs at 121 hPa, as discussed below.

Upon detailed inspection of the Aura MLS WV data and comparisons with frostpoint data, we have identified 4 common problematic profile shapes that occur in Aura MLS data, primarily at high latitudes. These features can

5 either be described as oscillations or "spikes" that occur at specific levels. The oscillations have a local minimum at 215 hPa and a local maximum at either 147 hPa or 121 hPa. The "spikes" are simply local maxima in water vapor mixing ratio (denoted here as q) at either 121 or 147 hPa. Quantitatively, we define these data artifacts as

121 hPa oscillation: $q_{261}>q_{215}<q_{177}$ & $q_{147}<q_{121}>q_{100}$

10 147 hPa oscillation: $q_{261}>q_{215}<q_{177}$ & $q_{178}<q_{147}>q_{121}$

121 hPa spike : $q_{147} < q_{121} > q_{100}$ & 121 hPa oscillation conditions not met

147 hPa spike: $q_{178} < q_{147} > q_{121}$ & 147 hPa oscillation conditions not met

Figure A3 shows the Aura MLS-FP comparison as in Fig. 3, but broken up by the four types of spikes/oscillations identified above. As can be seen in this figure, the two oscillation types contain extremely

dry-biased conditions at 215 hPa. In contrast, the local maxima at 121 hPa or 147 hPa are wet-biased relative to the frostpoint data.

As can be seen in Fig. A3, the Aura MLS data are biased at and below the level of the local maximum in the four types of features. For example, for Aura MLS profiles containing the 121 hPa oscillation (i.e., yellow dashed line in Fig. A3), the data at and below 121 hPa appear to be problematic, whereas the upper part of the

profile looks similar to Aura MLS profiles that don't contain an oscillation/spike feature. Because the Aura MLS profiles appear relatively unaffected at the higher levels when the data artifacts are present at the lower levels, we filter the data by simply removing the part of the profile below the local maximum (i.e., at 121 hPa or 147 hPa).

At some latitudes and during some seasons, this removes a significant fraction of Aura MLS data below 100 hPa.

Figure A4 illustrates the occurrence frequency of the four types of UTLS features found in Aura MLS data, binned into 10° latitude bins by month. As can be seen in this plot, the data artifacts occur in more than 50% of Aura MLS profiles for most of the year at latitudes poleward of 50° latitude in each hemisphere.

It is worth noting that Aura MLS retrievals with data artifacts mostly have the artifact present in their corresponding apriori profile. The data artifacts identified here may be due to an inherent difficulty in retrieving

low WV mixing ratios at high pressures with the Aura MLS instrument, as manifested in the apriori retrieval. Under these conditions, the dry continuum emission is the dominant absorber, and errors in the independent



tangent pressure/temperature retrievals that are used to estimate and remove the dry continuum emission could lead to "knock-on" effects in the retrieved WV. Indeed, the occurrence pattern of the data artifacts at 147 and 121 hPa (Fig. A4e-f) exhibit a pattern that looks to be related to temperatures in the UTLS (Fig. A4 g and h). In particular, the occurrence pattern at 147 hPa appears to correlate loosely with the temperature at 300 hPa

(compare Fig. A4e and A4g), whereas the pattern at 121 hPa correlates with the temperature at 150 hPa (Fig. A4f and A4h). Interestingly, the correlation patterns are reversed. The 147 hPa artifact is anti-correlated with temperature; lower temperatures at 300 hPa correlate with a higher frequency of occurrence of the 147 hPa artifact. In contrast, the 121 hPa artifact is positively correlated with temperature. The robustness and reasons for these correlations is unknown, and warrants further study.

Finally, it is worth noting that the identification of a dry bias in the Aura MLS UTLS WV data has implications for the interpretation of disagreements between the Aura MLS UTLS WV data and reanalyses, as presented in (Jiang et al., 2015). They found that UTLS WV in several of the modern reanalyses was ~150% higher than for Aura MLS, and interpreted the results as being indicative of a high bias in the reanalyses. Based on the frostpoint comparisons presented here and in other validation work (Read et al., 2007; Vömel et al., 2007a), we

find that the opposite may be the case, at least for high latitudes.

**Appendix B: Uncertainties in the SWOOSH products**

For each satellite source record monthly mean value and combined monthly mean value in SWOOSH, there is a corresponding uncertainty estimate and a standard deviation stored. This appendix provides a description of the source record uncertainty estimates and how the source record standard deviation and uncertainty estimates are

combined.

**B.1 Uncertainty estimates**

Before computing the monthly mean value for a given satellite in a given latitude/height/month bin, each "raw" satellite measurement within the bin is corrected. This is done by first interpolating the offset values (which are stored in 10° latitude bins and vary only with height/latitude) to the latitude of the satellite measurements. Then,

the offsets are added to the raw measurements, i.e.

$$q_{corrected_{kn}} = q_{raw_{kn}} + q_{offset_{kn}} \tag{B1}$$

where $q_{raw_{kn}}$ is the $n^{th}$ uncorrected satellite measurement from satellite $k$ within the (latitude/height/month) bin, $q_{offset_{kn}}$ is the additive offset interpolated from the 10° grid (Figs. 5-6) to the latitude of $q_{raw_{kn}}$, and



$q_{corrected_{kn}}$ is the corrected value. These $N_k$ corrected measurements are combined to form the monthly mean value $\bar{q}_k$ for each satellite,

$$\bar{q}_k = \frac{\sum_{n=1}^{N_k} q_{corrected_{kn}}}{N_k} = \frac{\sum_{n=1}^{N_k} q_{raw_{kn}}}{N_k} + \frac{\sum_{n=1}^{N_k} q_{offset_{kn}}}{N_k} \qquad (B2)$$

Given the satellite retrieval precision estimates $\sigma_{q_{raw_{kn}}}$ provided with every measurement by the satellite teams and the uncertainty in the offset $\sigma_{q_{offset_{kn}}}$ (described in section 4.1), propagation of errors gives an uncertainty in the monthly mean value of

$$\sigma_{\bar{q}_k} = \sqrt{\frac{1}{N_k^2}\sum_{n=1}^{N_k} \sigma_{q_{raw_{kn}}}^2 + \frac{1}{N_k^2}\sum_{n=1}^{N_k} \sigma_{q_{offset_{kn}}}^2} = \frac{\sigma_{rmss_k}}{\sqrt{N_k}} \qquad (B3)$$

where $\sigma_{rmss_k}$ is the root-mean-sum-square error of the N profiles in the bin, given by

$$\sigma_{rmss_k} = \sqrt{\frac{\sum_{n=1}^{N}\left(\sigma_{q_{raw_{kn}}}^2 + \sigma_{q_{offset_{kn}}}^2\right)}{N_k}} \qquad (B4)$$

The $\sigma_{rmss_k}$ values can be thought of as an average uncertainty for $N_k$ uncertainty estimates. In SWOOSH, the $\sigma_{rmss_k}$ values for each satellite are stored (e.g., `mlsh2ormssunc`). Analogously, $\sigma_{\bar{q}_k}$ can be thought of as the standard error of the mean for the combined SWOOSH product.

After creating the source record monthly mean for each satellite (Eq. B2), they are combined to form the "combined" monthly-mean product (e.g., variable `combinedh2oq`), using a weighted mean based on the number of observations $N_k$. The combined average of K satellites is then given by

$$\bar{q} = \sum_{k=1}^{K} \bar{q}_k \frac{N_k}{N_{tot}} \qquad (B5)$$

where $N_{tot} = \sum_{k=1}^{K} N_k$

and the uncertainty on the combined mean value is

$$\sigma_{\bar{q}}^2 = \frac{1}{N_{tot}^2}\sum_{k=1}^{K} \sigma_{\bar{q}_k}^2 N_k^2 \qquad (B6)$$

By substituting (B3) from above, this uncertainty can be simplified to

$$\sigma_{\bar{q}} = \frac{\sigma_{rmss}}{\sqrt{N_{tot}}} \qquad (B7)$$

with

$$\sigma_{rmss} = \sqrt{\sum_{k=1}^{K} \sigma_{rmss_k}^2 \frac{N_k}{N_{tot}}} \qquad (B8)$$



In this formulation, $\sigma_{rmss}$ (the combined uncertainty, stored as, e.g., `combinedh2ormssunc`) is expressed as the combination (in quadrature) of the individual satellite $\sigma_{rmss_k}$ values, weighted by the number of measurements $N_k$ for a given satellite, in the same way as the combined monthly mean (Eq. B5).

**B.2. Combined standard deviation**

In this section, we describe how the individual source record standard deviations are combined to form the standard deviation for the combined data product. In its most simple form, the sample variance (i.e., the standard deviation squared) for the combined product, $s^2$, is

$$s^2 = \frac{1}{N_{tot}-1}\sum_{l=1}^{N_{tot}}(q_l - \bar{q})^2 \tag{B9}$$

where $q_l$ is simply the collection of all the individual satellite measurements in the given lat/height/month bin.

Since the SWOOSH processing involves gridding each of the K satellites separately and then combining them, it is not possible to compute $s^2$ directly, as in the above equation.

Instead, we seek an expression for $s^2$ in terms of the gridded source record means ($\bar{q}_k$) and variances ($s_k^2$). The source record variance is

$$s_k^2 = \frac{1}{(N_k-1)}\sum_{n=1}^{N_k}(q_{kn} - \bar{q}_k)^2 \tag{B10}$$

with $q_{kn}$ being the corrected source record data (i.e, $q_{corrected_{kn}}$).

To begin recasting Eq. (B9) in terms of the gridded source records, we re-write it as

$$(N_{tot}-1)s^2 = \sum_{l=1}^{N_{tot}}q_l^2 - 2\bar{q}\sum_{l=1}^{N_{tot}}q_l + N_{tot}\bar{q}^2 \tag{B11}$$

Since by definition $\bar{q} = \frac{1}{N_{tot}}\sum_{l=1}^{N_{tot}}q_l$, this simplifies to

$$(N_{tot}-1)s^2 = \sum_{l=1}^{N_{tot}}q_l^2 - N_{tot}\bar{q}^2 \tag{B12}$$

Through a similar re-arrangement of Eq. (B10), we get

$$(N_k-1)s_k^2 = \sum_{n=1}^{N_k}q_{kn}^2 - N_k\bar{q}_k^2 \tag{B13}$$

And since

$$\sum_{l=1}^{N_{tot}}q_l^2 = \sum_{k=1}^{K}\sum_{n=1}^{N_k}q_{kn}^2 \tag{B14}$$

we can re-write Eq. (B14) by substituting Eq. (B13)

$$\sum_{l=1}^{N_{tot}}q_l^2 = \sum_{k=1}^{K}(N_k-1)s_k^2 + \sum_{k=1}^{K}N_k\bar{q}_k^2 \tag{B15}$$

This expression can then be substituted into Eq. (B12), and re-arranged to give the variance for the combined product





$$s^2 = \frac{1}{N_{tot}-1}\sum_{k=1}^{K}(N_k-1)s_k^2 + \frac{1}{N_{tot}-1}\sum_{k=1}^{K}N_k\bar{q}_k^2 - \frac{N_{tot}}{N_{tot}-1}\bar{q}^2 \qquad (B16)$$

Using this equation, the combined standard deviation is computed directly from the source variances ($s_k^2$), source means ($\bar{q}_k$), and the combined mean value ($\bar{q}$).

For the case where there are a sufficiently large number of data points such that $N_{tot} \approx N_{tot} - 1$ and $N_k \approx N_k - 1$, Eq. (B16) reduces to

$$s^2 \approx \frac{1}{N_{tot}}\sum_{k=1}^{K}N_k s_k^2 + \frac{1}{N_{tot}}\sum_{k=1}^{K}N_k\bar{q}_k^2 - \bar{q}^2 \qquad (B17)$$

In this approximation, it is easy to see the meaning of the terms in Eq. (B16). The first term represents the contribution to the combined variance from the source record variances, and is simply a weighted mean of the source variances (again, with weighting based on the number of measurements). The second and third terms represent a contribution to the variance from the "spread" of the source means $\bar{q}_k$ about the combined mean $\bar{q}$.

That the last two terms of Eq. (B17) represent a spread of $\bar{q}_k$ around $\bar{q}$ is easiest to see in the special case where the number of data points for each source record ($N_{source}$) is the same (i.e., $N_{source} * K = N_{tot}$ and $\bar{q} = \frac{1}{K}\sum_{k=1}^{K}\bar{q}_k$). In this case, the second and third terms are equivalent to the variance of $\bar{q}_k$, i.e.,

$$s_{\bar{q}_k}^2 = \langle\bar{q}_k^2\rangle - \bar{q}^2 = \frac{1}{K}\sum_{k=1}^{K}\bar{q}_k^2 - \bar{q}^2 = \frac{N_{source}}{N_{tot}}\sum_{k=1}^{K}\bar{q}_k^2 - \bar{q}^2 \qquad (B18)$$

and Eq. (B17) can be re-written as

$$s^2 \approx \frac{1}{N_{tot}}\sum_{k=1}^{K}N_k s_k^2 + s_{\bar{q}_k}^2 \qquad (B19)$$

Thus, in this special case, the combined variance is a sum of the "spread" of the source data around each source mean (first term in Eq. (B19)) and the "spread" of the source means about the combined mean (second term in Eq. (B19)).

Another special case for Eq. (B16) occurs when the source monthly means are equal to each other (i.e., $\bar{q}_k = \bar{q}$). In this case, the second and third terms in Eq. (B16) cancel, and the variance in the combined product is just the weighted mean of the $s_k^2$ in the first term. This special case is most applicable in the SWOOSH database, because the source records have been corrected such that $\bar{q}_k \approx \bar{q}$. Nevertheless, the combined standard deviation stored in SWOOSH is computed from the full expression shown in Eq. (B16). Data are stored with names like, e.g., `combinedh2ostddev`.



**Acknowledgements**

The authors thank the satellite data teams for their efforts in retrieval, quality control, documentation, and dissemination of their data. The authors thank the balloon sounding teams for providing their data. We also thank Roy Miller for assistance in downloading MERRA data and for general data management help. Frost

5    point hygrometer soundings at Boulder, Hilo, and Lauder are supported by NASA Award NNX13AK55G, the NOAA Climate Program Office, and the US Global Climate Observing System. This study was supported in part by NOAA's Climate Program Office. The UARS and Aura MLS data sets were produced at the Jet Propulsion Laboratory, California Institute of Technology under contract with the National Aeronautics and Space Administration. The authors would also like to thank Ellis Remsberg, John Anderson, Ray Wang, and

10    Lucien Froidevaux for helpful discussions and advice.



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



**Figures**

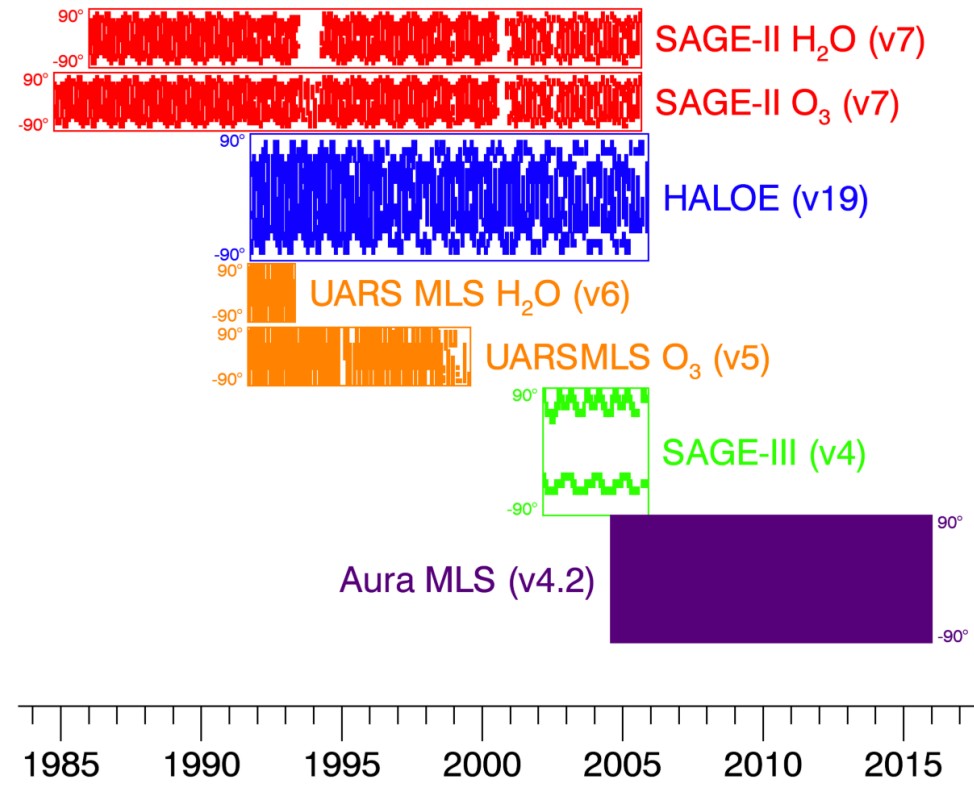

**Figure 1:** Temporal and latitudinal coverage of satellite data used as input to SWOOSH. The boxes surrounding each data set span 90°S – 90°N in the vertical. Data are filled for each month and 10° latitude band containing valid data. Where significant coverage differences exist for WV and $O_3$ for a given satellite, the coverage is plotted separately.



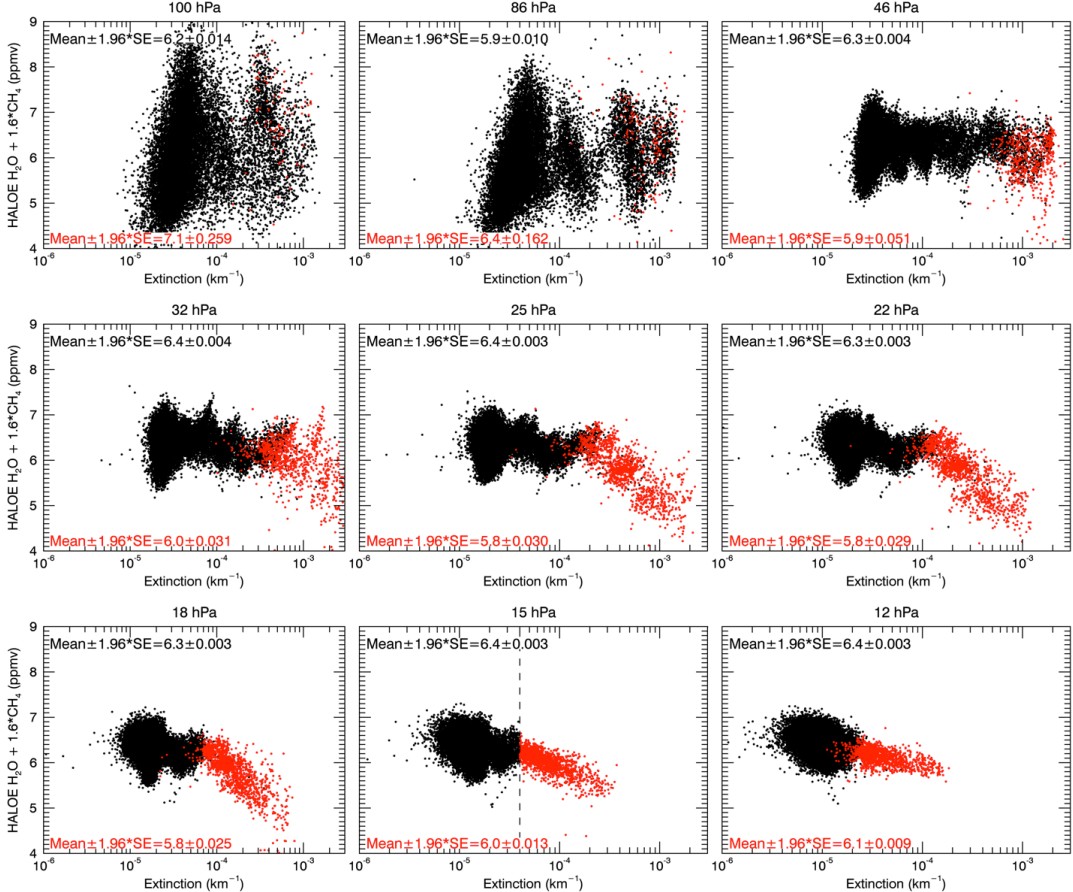

**Figure 2: HALOE "total water" (H$_2$O + 1.6*CH$_4$) versus aerosol extinction in the HALOE NO channel for tropical (30°S – 30°N) data, segregated by events with low extinction (< 4 x 10$^{-5}$ km$^{-1}$, black) and high extinction (> 4 x 10$^{-5}$ km$^{-1}$, red) at 15 hPa. The mean ± 1.96 * standard error of the mean (i.e., the 95% confidence interval) is given for each level and extinction category.**



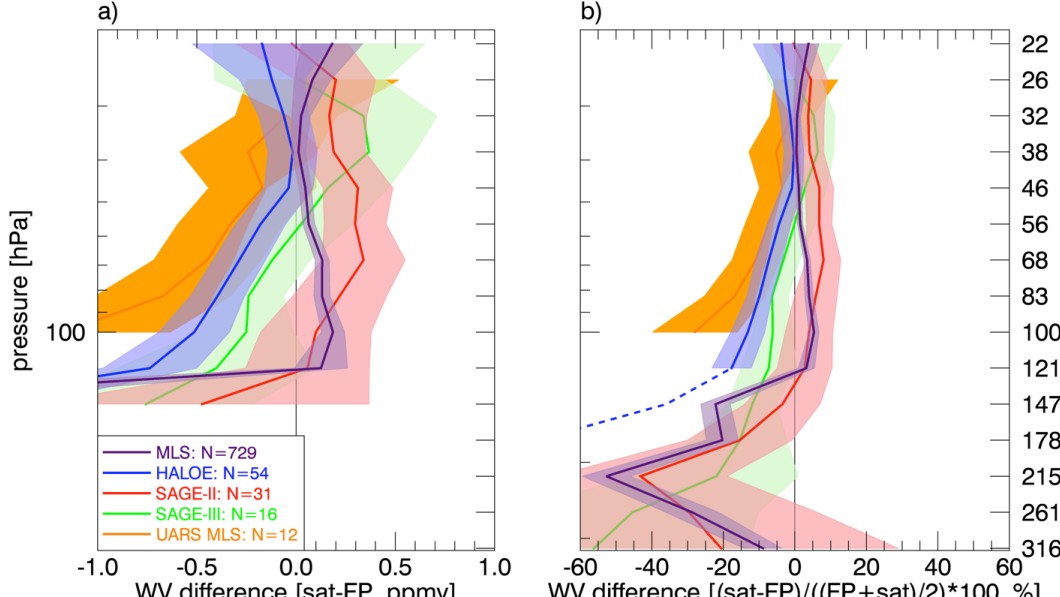

**Figure 3:** The difference as a function of height between matched satellite and balloon frostpoint hygrometer water vapor data, expressed in (a) as a mixing ratio difference and in (b) as a percent difference between the mean value at each level (see discussion in Sect. 3.1). The number of matches at 82 hPa is shown in the legend, and the mean difference (solid) and ±2 standard error ($2\sigma/\sqrt{N}$) range (shaded) are shown at each level. The dashed blue line shows the HALOE data that are excluded from SWOOSH, at and below 147 hPa.



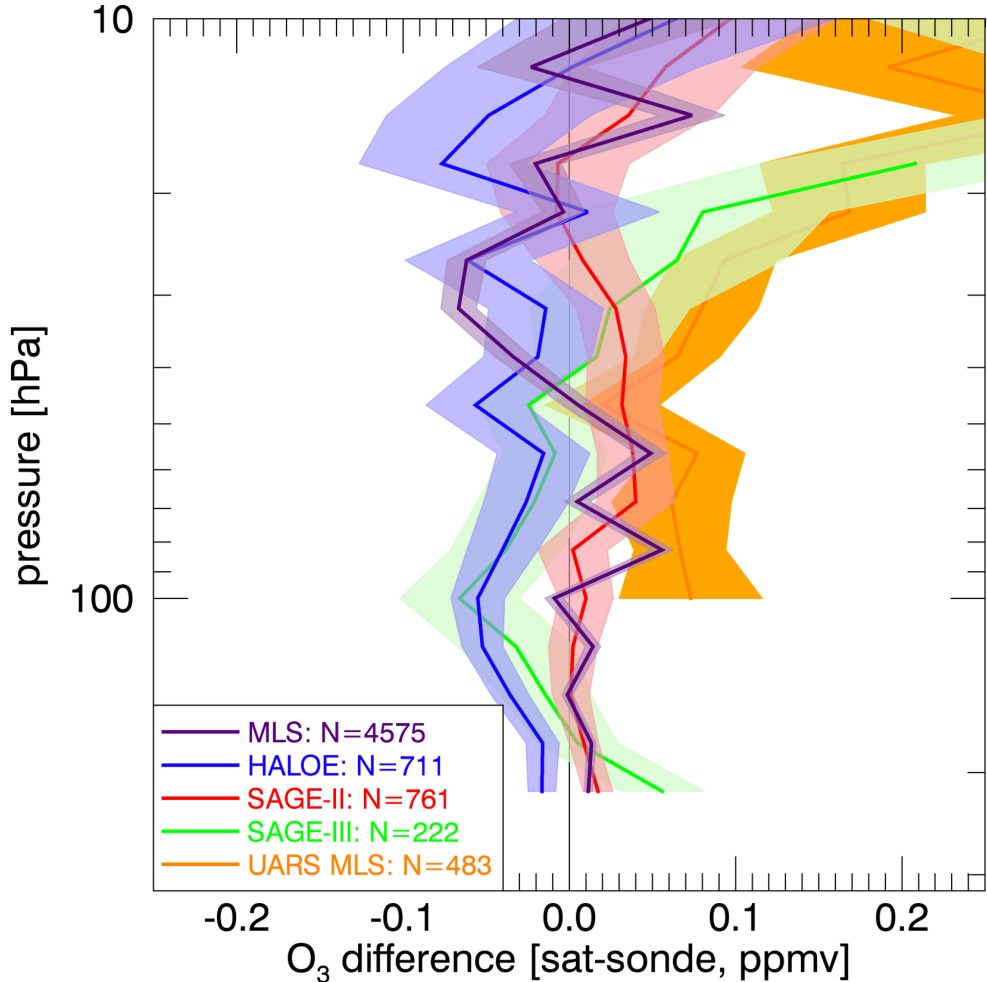

**Figure 4: The difference as a function of height between matched satellite and ozonesondes observations. The number of matches at 82 hPa is shown in the legend, and the mean difference (solid) and ±2 standard error ($2\sigma/\sqrt{N}$) range (shaded) are shown at each level.**



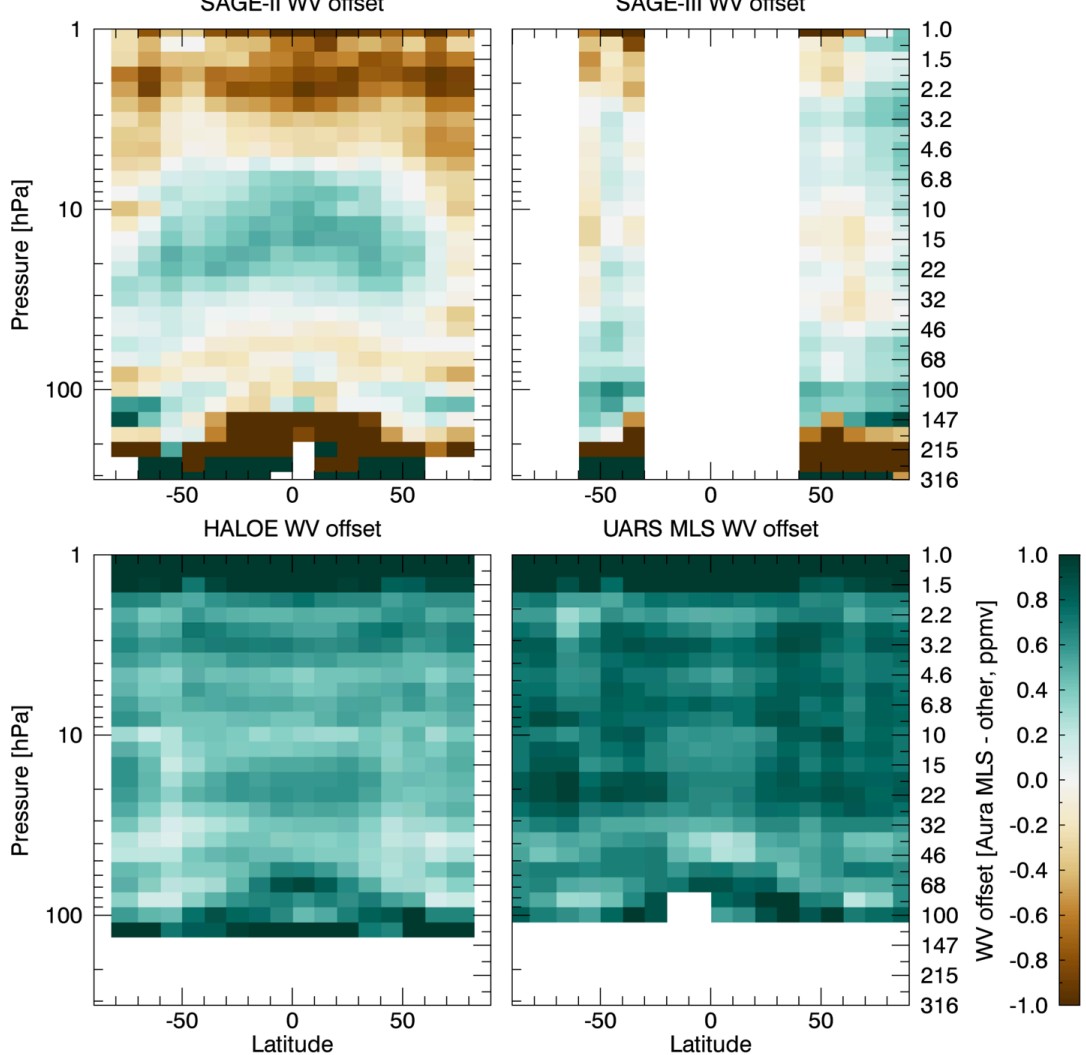

**Figure 5:** WV offsets relative to Aura MLS for satellite data sets used in SWOOSH. WV offsets are defined as the mean of MLS minus the given dataset. Offsets are computed on the MLS vertical grid in 10° latitude bands.





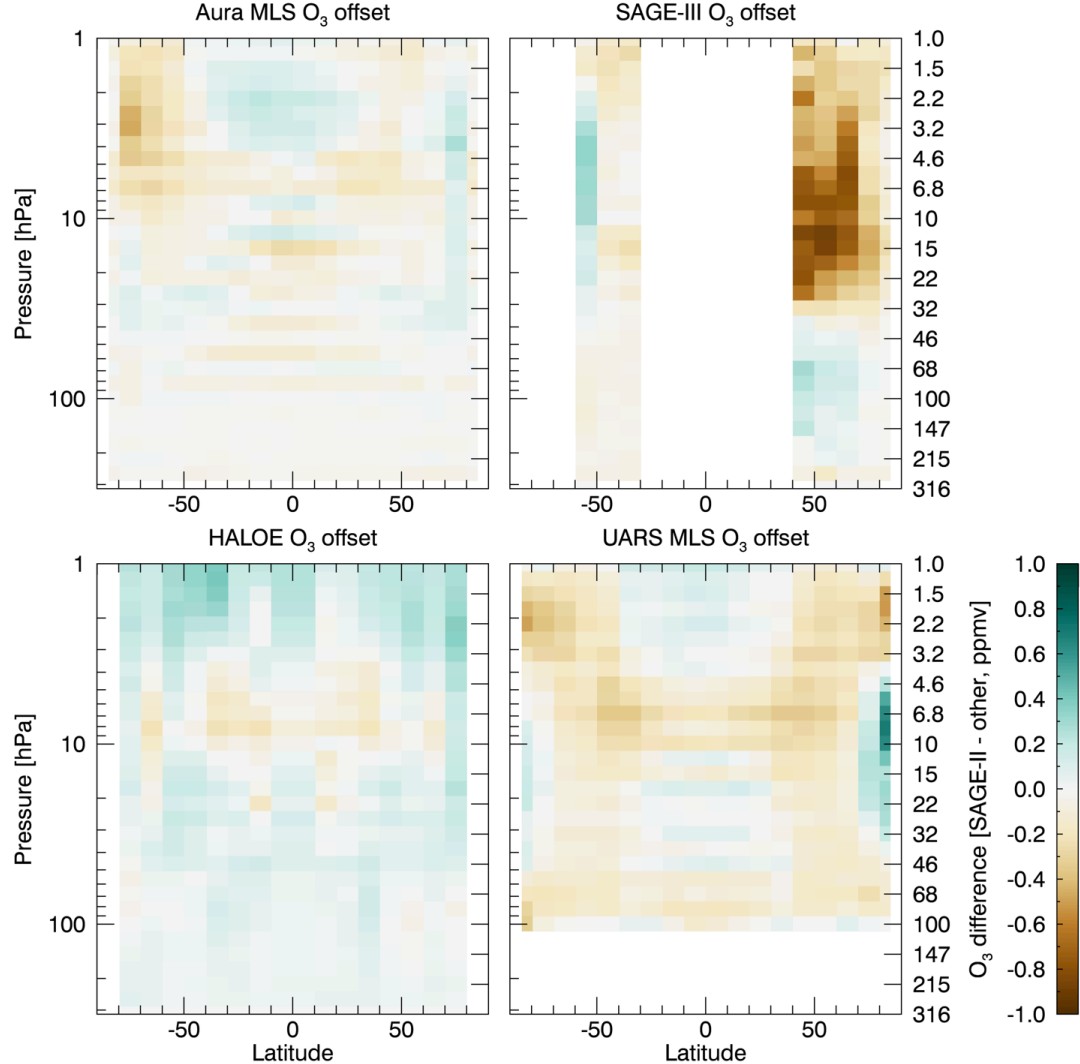

Figure 6: Same as Fig. 5, but for O₃ offsets, which are relative to the SAGE-II data.



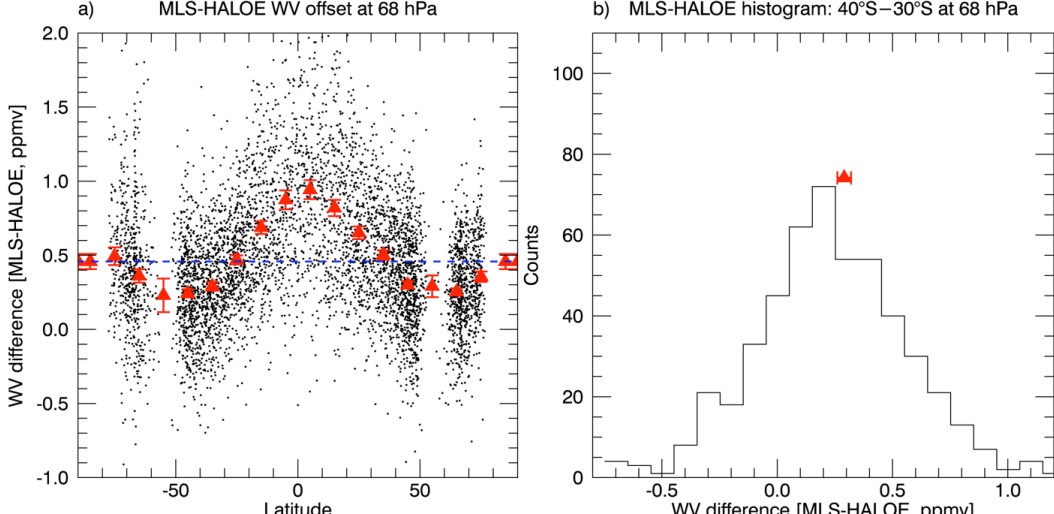

**Figure 7: Example of WV offset adjustment for HALOE at 68 hPa. (a) Matched MLS/HALOE pairs (dots), the 10° latitude binned means (red filled triangles) with error bars showing the offset uncertainties (95% confidence interval). The mean (over all latitudes) is shown as a horizontal blue dashed line. (b) The histogram of MLS-HALOE differences at 68 hPa for the 40°S - 30°S latitude bin. The offset uncertainty for this bin is shown as a horizontal red bar.**

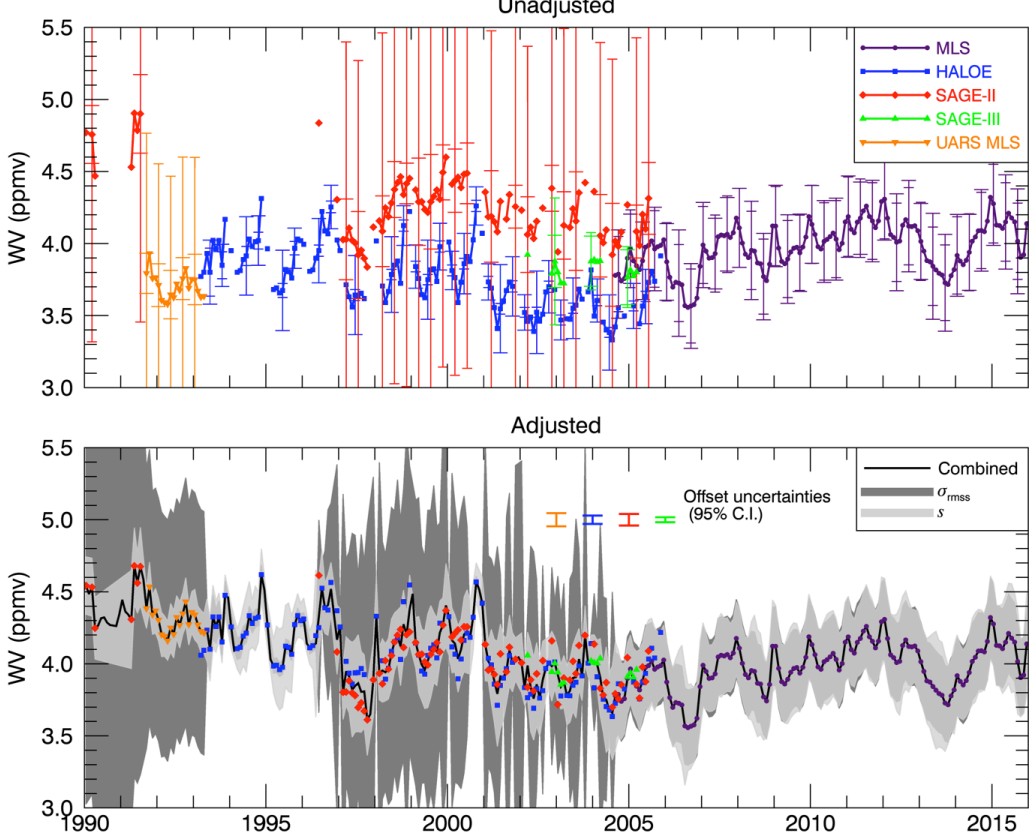

Figure 8: (Top) The uncorrected source water vapor timeseries in the 30° - 40°S latitude band at 68 hPa, along with the source standard deviation (wide error bars) and root-mean-sum-of-squares (RMSS) uncertainty values ($\sigma_{rmss_k}$, narrow error bars). (Bottom) The offset-corrected source measurements along with the combined ("anomaly filled") product. The lighter and darker gray shaded regions show the combined RMMS uncertainty ($\sigma_{rmss}$, narrow error bars) and the combined standard deviation ($s$), respectively. The vertical errorbars in the lower panel show the 95% confidence interval of the offset uncertainties for the 30° - 40°S latitude band at 68 hPa.



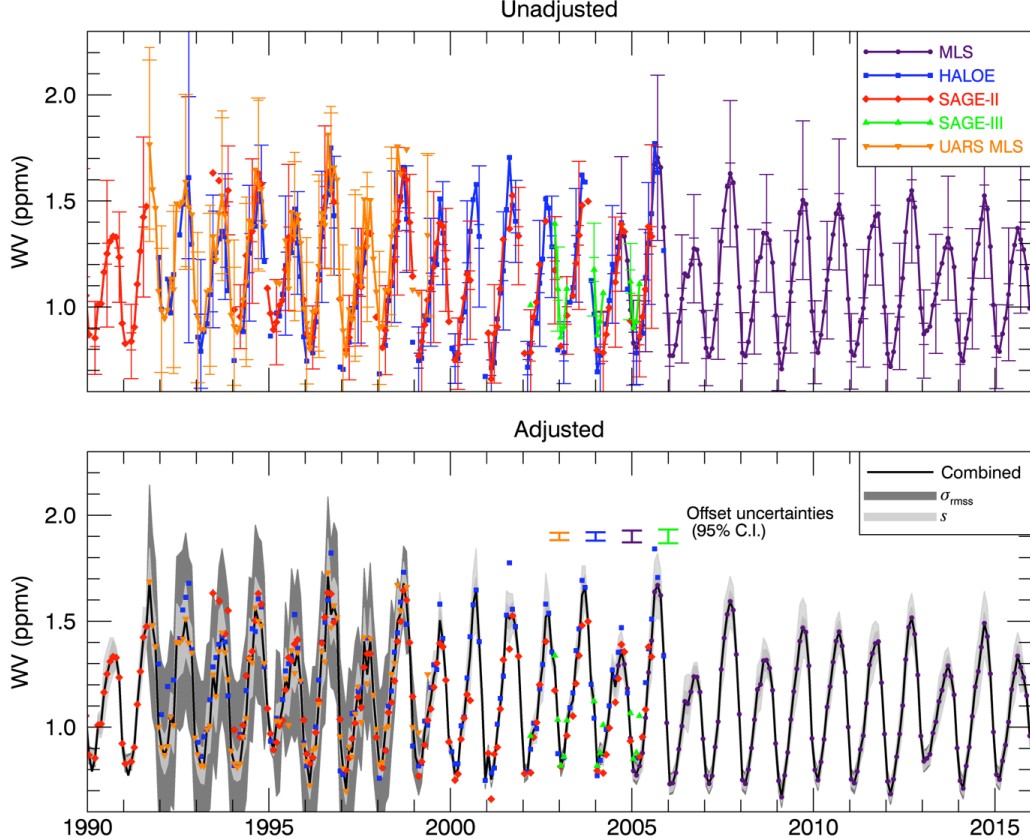

**Figure 9: Same as in Fig. 8, but for ozone.**





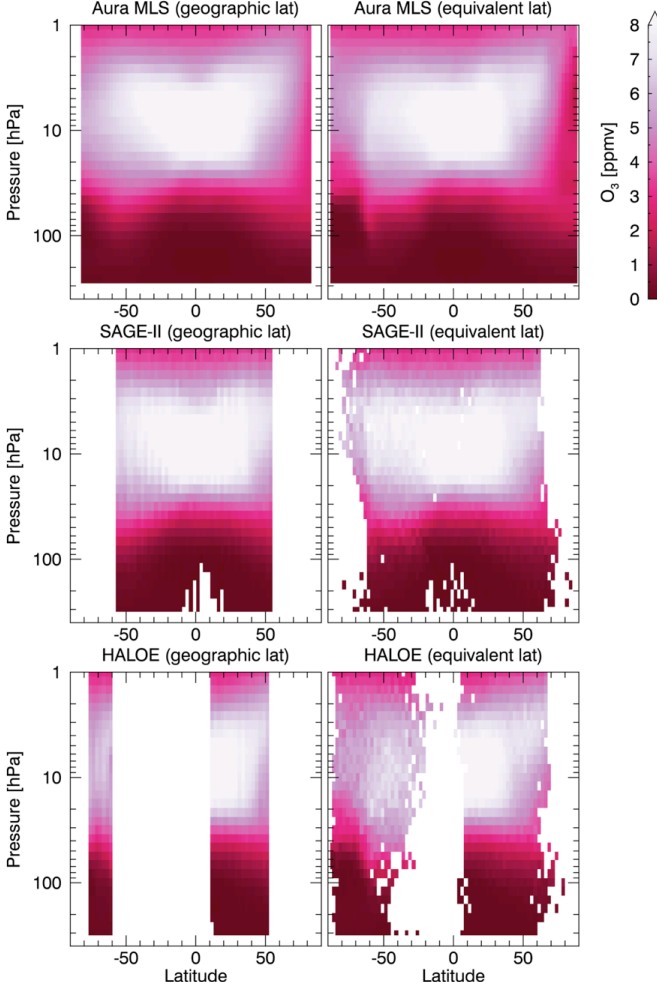

**Figure 10: Ozone height vs. latitude plots on geographic latitude (left column) and equivalent latitude grids (right column), for October 2004. Increased data coverage and increased depth of Antarctic ozone depletion is apparent in the equivalent latitude gridded data.**



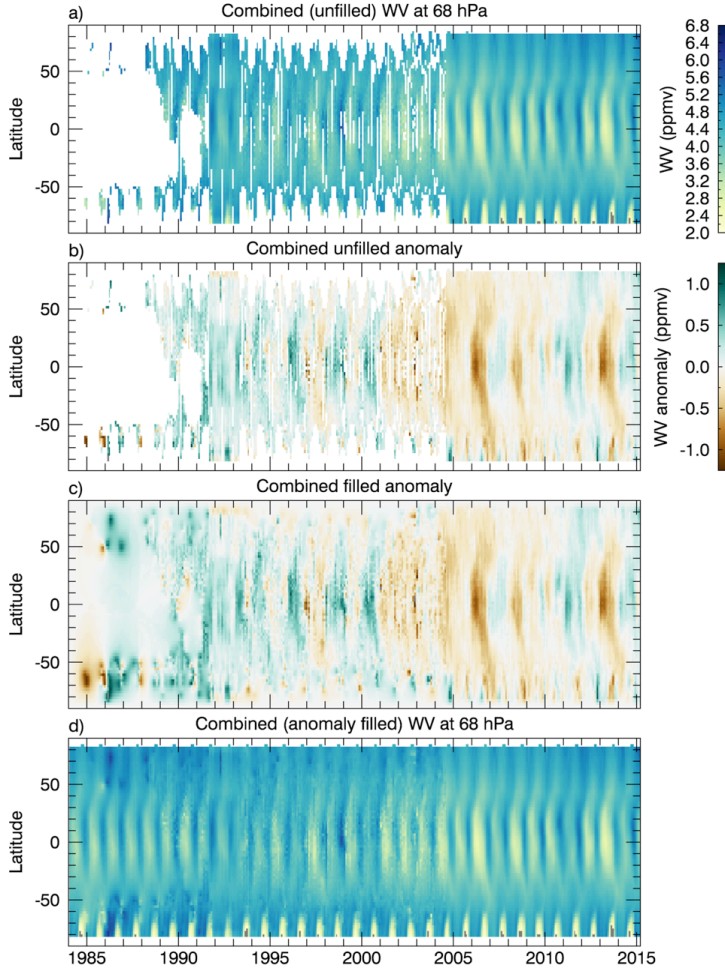

Figure 11: (a) Combined (unfilled) latitude vs. time cross-section of WV at 68 hPa. (b) The anomaly of a), defined as a) minus the seasonal cycle at each latitude. (c) A filled version of b), using radial basis function interpolation. (d) The combined (anomaly filled) product, which is constructed by adding the (latitude dependent) seasonal cycle to the filled anomaly c).



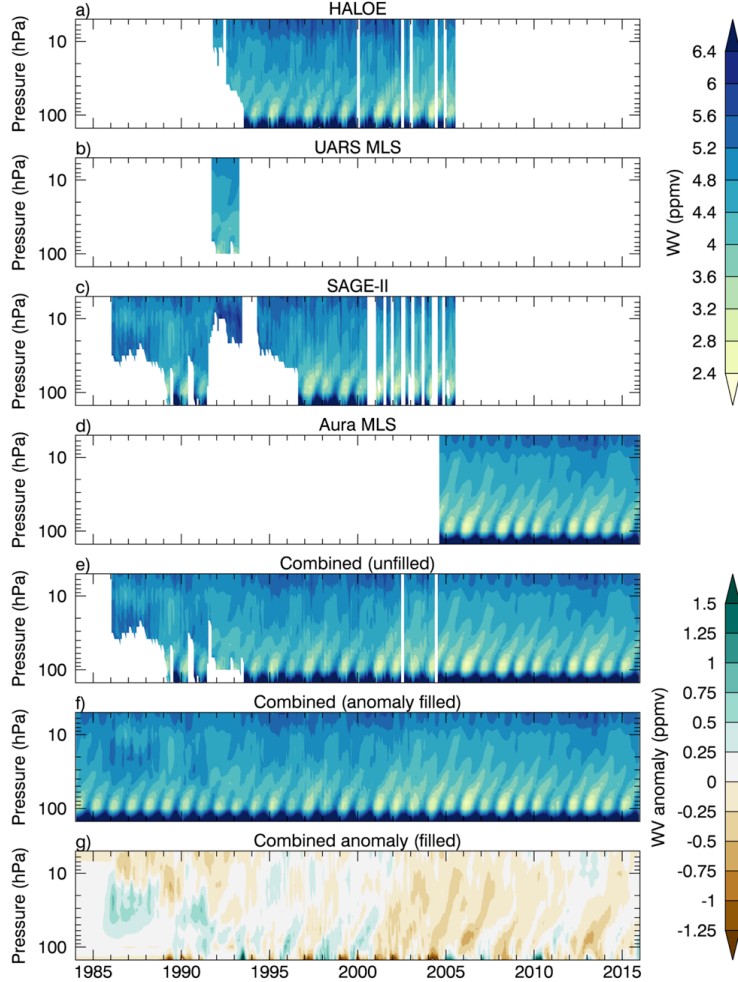

**Figure 12: The tropical average (30°S – 30°N) water vapor concentration as a function of height and time, which is commonly referred to as the "tropical tape recorder", for each satellite data set in SWOOSH (a-d), as well as the two combined products (e-f) and the combined (filled) anomaly product (g). SAGE-III data are nonexistent in the tropics, and therefore not included.**





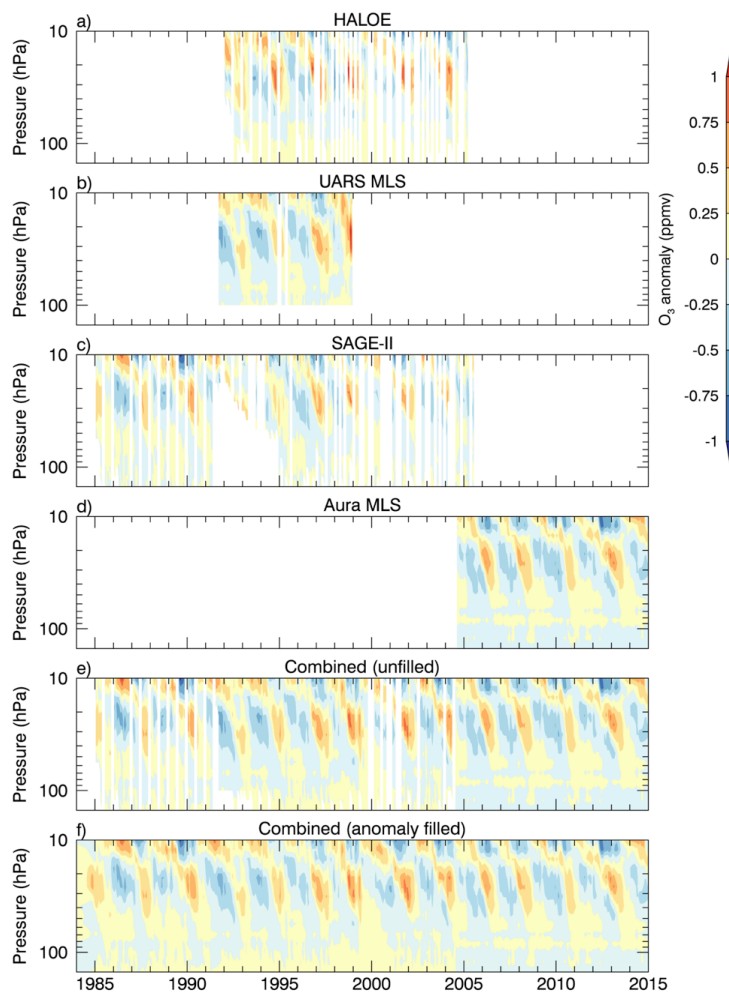

**Figure 13: The tropical average (10°S – 10°N) ozone concentration anomaly as a function of height and time for each satellite data set in SWOOSH (a-d), as well as the two combined products (e-f).**



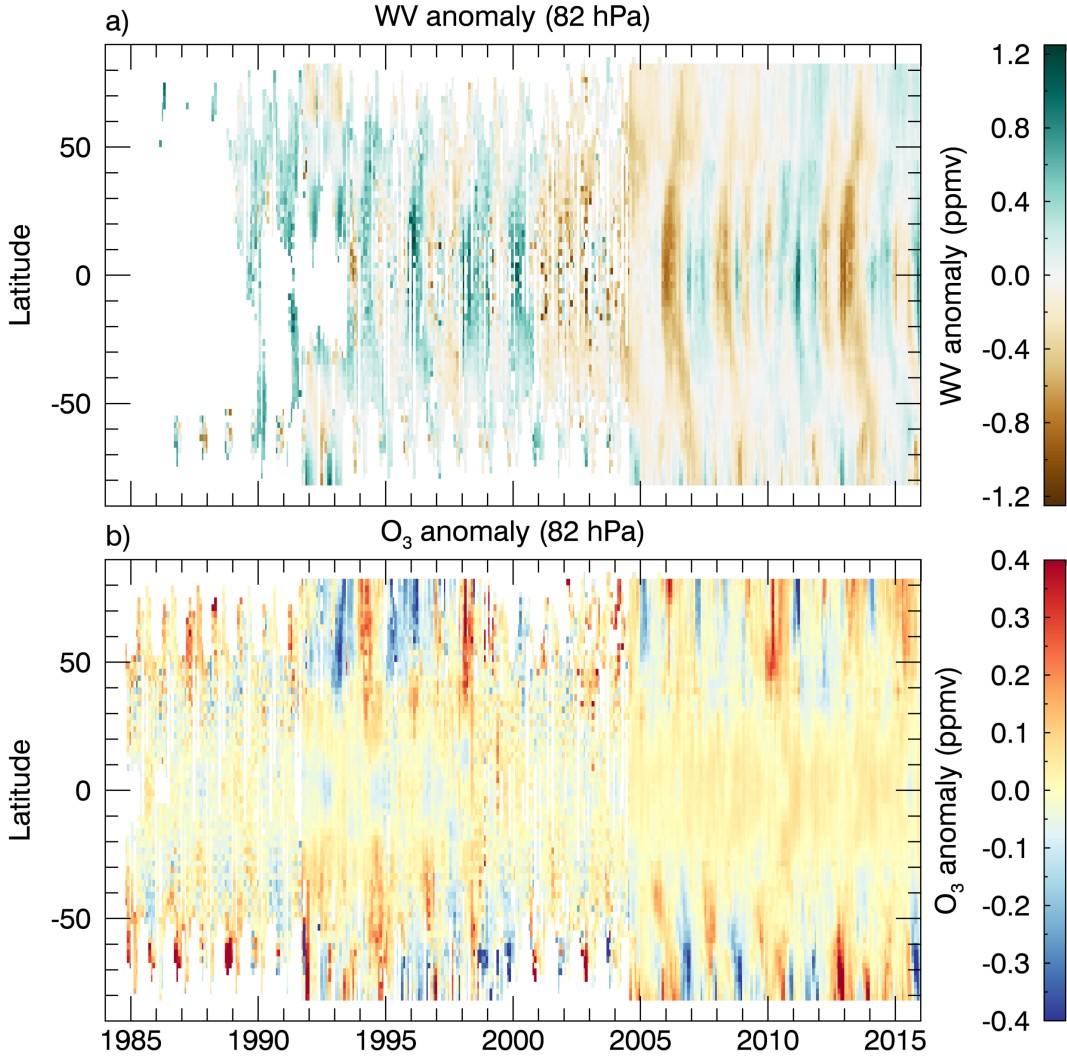

Figure 14: SWOOSH combined product water vapor and ozone anomalies (unfilled) at 82 hPa.



**Tables**

**TABLE 1. Overview of Satellite data included in SWOOSH**

| Instrument | Data Version | Start Date | End Date | $H_2O$ Vertical Resolution (km) | $O_3$ Vertical Resolution (km) | $H_2O$ Vertical Range | $O_3$ Vertical Range |
|---|---|---|---|---|---|---|---|
| SAGE II | 7 ($O_3$) 7 ($H_2O$) | 1984-10 1986-01 | 2005-08 2005-08 | $1^1$ | $1^1$ | 0.5-50 km[2] | 0.5-70 km[2] |
| UARS HALOE | 19 | 1991-10 | 2005-11 | 2 | 2.3 | 316 – 0.002 hPa | 584 – 0.0004 hPa |
| UARS MLS | 5 ($O_3$) 6 ($H_2O$) | 1991-10 ($H_2O$) 1991-09 ($O_3$) | 1993-04 2005-11 | $3\text{-}4^3$ | $3.5\text{-}5^4$ | 100 – 0.17 hPa | 100 – 0.22 hPa |
| SAGE III | 4 | 2002-02 | 2005-11 | 1.5 | 1 | 0.5 – 100 km[2] 8 – 49 km[5] 384 – 0.45 hPa[5] | 0.5 – 100 km[2] 1 – 82 km[5] 905 – 0.007 hPa[5] |
| Aura MLS | 4.2 | 2004-08 | present | 2.5-3.5 | 2.5-3.5 | 316 – 0.002 hPa | 261 – 0.02 hPa |

5    [1]: FWHM of triangular smoothing applied to balloon data for intercomparisons.
[2]: retrieval grid
[3]: 3.5 km FWHM used for UARS MLS WV smoothing
[4]: 4 km FWHM used for UARS MLS $O_3$ smoothing
[5]: range of valid values from raw data, before any screening



**TABLE 2. Frostpoint hygrometer stations used in satellite water vapor intercomparison**

| Station | Latitude | Longitude | # Soundings | Period |
|---|---|---|---|---|
| Alert (CAN) | 82 | -61.5 | 6 | 1989-1991 |
| Ny Alesund (NOR) | 78.9 | 11.9 | 6 | 2002-2004 |
| Thule (GRL) | 77.5 | -69 | 4 | 1994-1995 |
| Kiruna (SWE) | 67.8 | 20.2 | 19 | 1991-2003 |
| Sodankyla (FIN) | 67.2 | 26.4 | 114 | 1996-2012 |
| Fairbanks, AK (USA) | 64.8 | -147.7 | 4 | 1985-1997 |
| Keflavik (ISL) | 64 | 20.7 | 2 | 1994 |
| Lindenberg (GER) | 52.2 | 14.1 | 77 | 2006-2012 |
| Laramie, WY (USA) | 41.3 | -105.6 | 5 | 1983-1989 |
| Boulder (USA) | 40 | -105.2 | 397 | 1980-2014 |
| Beltsville, MD (USA) | 39 | -76.9 | 37 | 2006-2011 |
| Washington D.C. (USA) | 38.9 | -77 | 129 | 1964-1980 |
| Crows Landing, CA (USA) | 37.4 | -121.2 | 3 | 1993 |
| Lamont, OK (USA) | 36.6 | -97.5 | 12 | 2003 |
| Edwards AFB, CA (USA) | 34.9 | -117.9 | 4 | 1991 |
| Dagett, CA (USA) | 34.8 | -117 | 1 | 1992 |
| Huntsville, AL (USA) | 34.7 | -86.7 | 2 | 2002 |
| Fort Sumner, NM (USA) | 34.5 | -104.3 | 10 | 1996-2004 |
| Wrightwood, CA (USA) | 34.4 | -117.7 | 40 | 2006-2009 |
| Midland, TX (USA) | 31.9 | -102.2 | 1 | 2004 |
| Palestine, TX (USA) | 31.8 | -95.6 | 8 | 1981-1985 |
| Lhasa (CHN) | 29.7 | 91.1 | 20 | 2010-2012 |
| Kunming (CHN) | 25 | 102.7 | 36 | 2009-2012 |
| Tengchong (CHN) | 25 | 98.5 | 12 | 2010 |
| Yangjiang (CHN) | 21.9 | 112 | 12 | 2010 |
| Ha Noi (VNM) | 21 | 105.8 | 23 | 2007-2011 |
| Hilo, HI (USA) | 19.7 | -155.1 | 55 | 1991-2014 |
| Pago Pago (ASM) | 14.3 | -170.7 | 5 | 1986-1988 |
| San Jose (CRI) | 10 | -84.1 | 167 | 2005-2014 |
| Tarawa (KIR) | 1.4 | 172.9 | 10 | 2005-2010 |
| Kototabang (IDN) | -0.2 | 100.3 | 9 | 2007-2008 |
| San Cristobal (ECU) | -0.9 | -89.6 | 47 | 1998-2007 |
| Biak (IDN) | -1.2 | 136.1 | 34 | 2006-2011 |
| Bandung (IDN) | -6.9 | 107.6 | 8 | 2003-2004 |
| Juazeiro do Norte (BRA) | -7.2 | -39.3 | 5 | 1997 |
| Watukosek (IDN) | -7.5 | 112.6 | 7 | 2001-2003 |
| R/V Mirai- Cindy | -8 | 80.5 | 39 | 2011 |
| Vickers Cruise | -9.4 | 160 | 14 | 1993 |
| La Reunion (REU) | -21.1 | 55.5 | 11 | 2005-2011 |
| Lauder (NZL) | -45 | 169.7 | 121 | 1992-2014 |



| McMurdo Station (ATA) | -77.8 | 166.7 | 31 | 1987-1999 |
|---|---|---|---|---|
| South Pole (ATA) | -90 | 0 | 22 | 1990-1994 |



**TABLE 3. Ozonesonde stations used in satellite ozone comparison**

| Station | Latitude | Longitude | # Soundings | Period |
|---|---|---|---|---|
| Alert (CAN) | 82.5 | -62.3 | 1028 | 1987-2011 |
| Resolute (CAN) | 74.7 | -95.0 | 885 | 1978-2011 |
| Uccle (BEL) | 50.8 | 4.4 | 2299 | 1996-2013 |
| Boulder (USA) | 40.0 | -105.3 | 698 | 1991-2015 |
| Wallops (USA) | 37.9 | -75.5 | 1779 | 1970-2013 |
| Hilo (USA) | 19.4 | -155.0 | 1717 | 1982-2013 |
| Natal (BRA) | -5.5 | -35.3 | 661 | 1979-2013 |
| Samoa (USA) | -14.2 | -170.6 | 992 | 1995-2013 |
| Lauder (NZL) | -45.0 | 169.7 | 1275 | 1986-2008 |
| Davis (ATA) | -68.6 | 78.0 | 270 | 2003-2013 |
| Neumeyer (ATA) | -70.7 | -8.3 | 1553 | 1992-2015 |



**TABLE 4. Available SWOOSH grids**

| Longitude | Latitude | Vertical type |
| --- | --- | --- |
| - | 2.5°, 5°, or 10° | Pressure |
| - | 2.5°, 5°, or 10° | Isentropic |
| 20° | 5° | Pressure |
| 30° | 10° | Pressure |



**Appendix Figures**

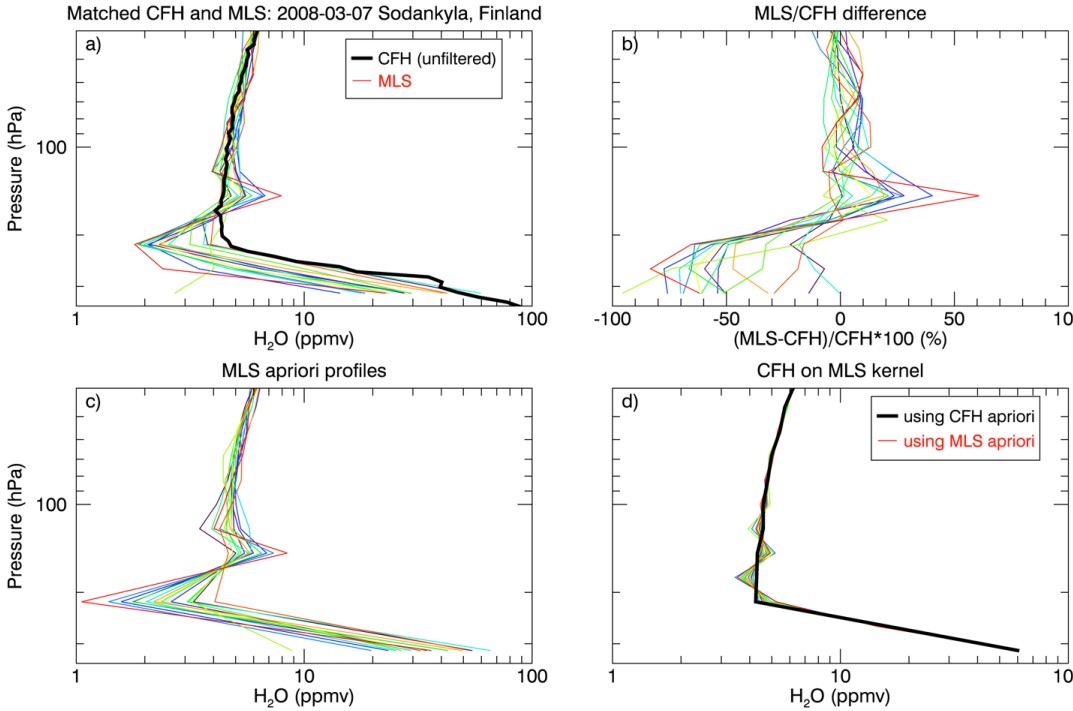

**Figure A1:** **(a) Cryogenic Frostpoint Hygrometer H$_2$O profile (black) taken from Sodankyla, Finland (67°N) on 2008-03-07, along with the 15 closest matched Aura Microwave Limb Sounder H$_2$O profiles (colored). (b) The difference between the matched MLS profiles and the CFH data. (c) The MLS a priori profiles corresponding to the MLS H$_2$O retrieval values shown in the upper left panel. (d) The CFH data averaged to the MLS resolution using different a priori profiles as input to the procedure described by Read et al. (2007). The black line is based on the CFH as the a priori, whereas the colored lines are the result of using the corresponding colored line from the lower left panel.**



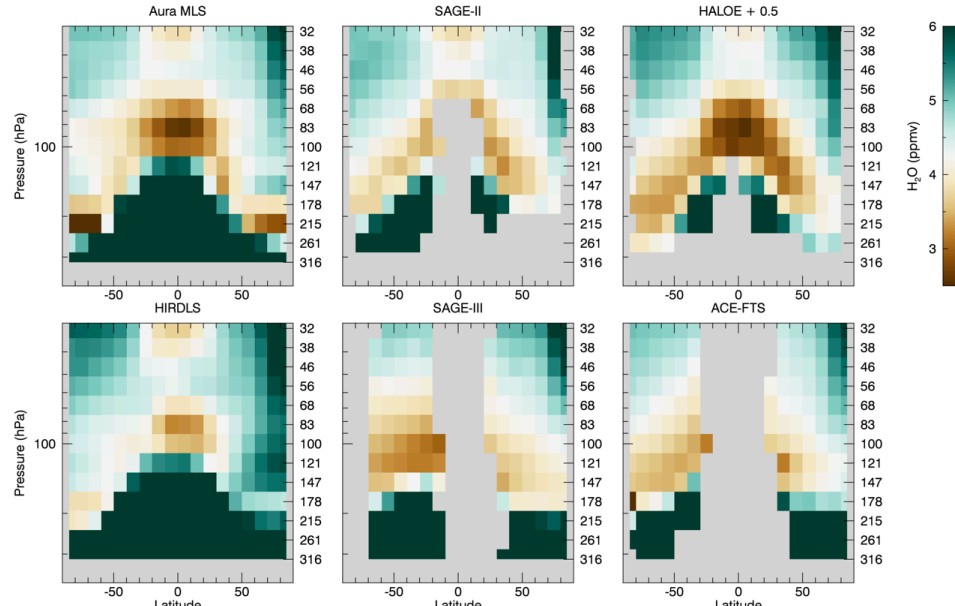

**Figure A2: Zonal-mean height vs. latitude cross-section of water vapor from six satellite instruments for the month of March, averaged from 2001 – 2009. Data are gridded on a 10° latitude grid using PV-based equivalent latitude from MERRA.**




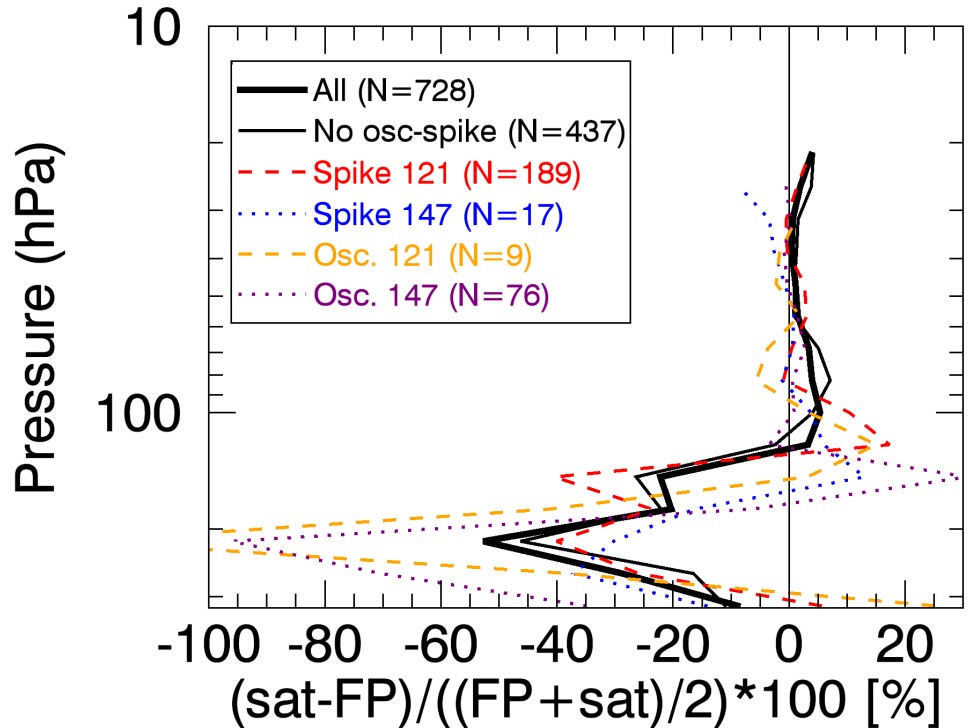

**Figure A3: Comparison between Aura MLS WV and frostpoint hygrometer sounding data, (similar to Fig. 3), separated by category of UTLS oscillation in the MLS data.**



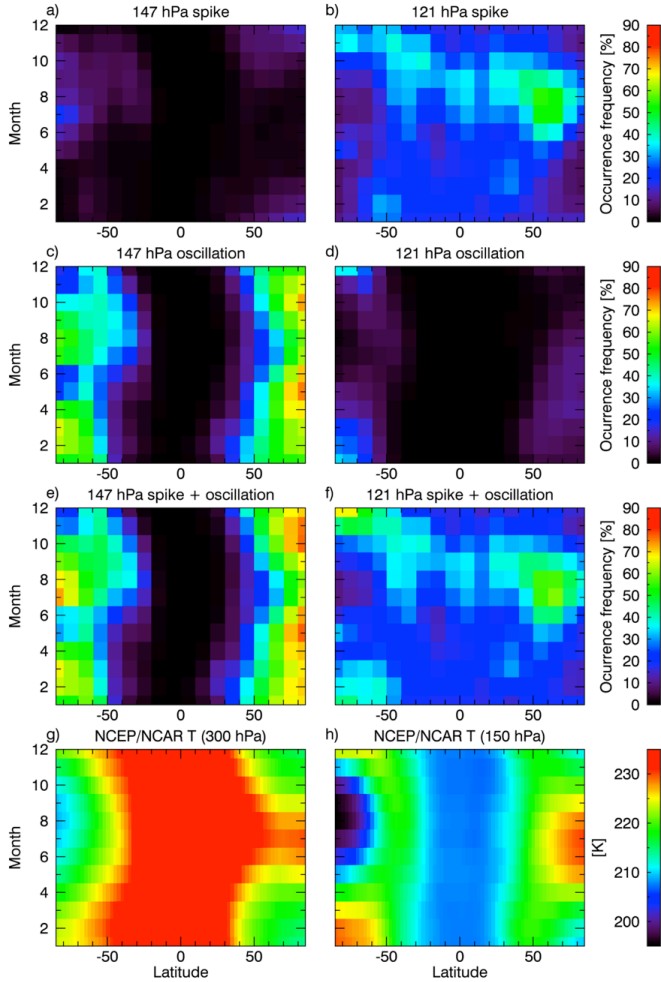

**Figure A4: (Top rows) Occurrence frequency statistics for the four types of Aura MLS UTLS WV oscillations identified. (Bottom row) NCEP/NCAR reanalysis 1981-2010 climatological mean zonal-mean temperature at 300 hPa and 150 hPa.**

