# Peer review of "The Stratospheric Water and Ozone Satellite Homogenized (SWOOSH) database: A long-term database for climate studies"

_Earth System Science Data, 2016_

## Referee Comment (RC1) · A. Dessler (Referee) · 15 Jun 2016

This paper describes a combined water vapor and ozone data set. This could be a widely used data set and the paper describes it well. I recommend publication after the authors consider my comments below.

My most important comment has to do with the construction of the combined anomaly data. Looking at earlier versions of SWOOSH, I remember that the different satellites had different shape seasonal cycles (at least in certain places), and this caused problems in the combined anomaly data. So I ended up doing my own merge by taking the anomaly time series for each individual satellite and merging those, which required subtracting the offset during their overlap periods. I presume this problem with the seasonal cycles is still an issue. I'm not recommending they change anything in how the data set is presently constructed, but I would suggest that the authors mention something about this as well as the fact that users of the data might want to create their own merged data set.

page 7, line 8: The conserved quantity in HALOE is between 1.7 and 1.9 ppmv $H_2O$/ppmv $CH_4$, as described by Dessler and Kim [Determination of the amount of water vapor entering the stratosphere based on HALOE data, J. Geophys. Res., 104, 30,605-30,607, 1999; see Table 1]. I don't know if that'll make a difference, but the authors should use the correct value.

Also, does "discard the profile below 15 hPa" mean altitudes below 15 hPa or pressures below 15 hPa?

page 10, line 5: In the calculation of the uncertainty, do the authors take autocorrelation into account? Because of temporal autocorrelation, the number of independent pieces of information is less than the number of months, and this needs to be accounted for. See, e.g., Santer et al. (2000), Statistical significance of trends and trend differences in layer-average atmospheric temperature time series, J. Geophys. Res., 105, 7337-7356, doi: 10.1029/1999jd901105.

page 10, line 26: Something is wrong with this sentence; the word "seven" doesn't make sense.

An aside: the authors spend a lot of effort matching satellite measurements with other satellites and sondes. These approaches have been done for decades, and I find them pretty unconvincing. The "match" criteria is set to get a large number of matches rather than because the criteria actually designate similar air masses. While I don't doubt the answers that the authors get are right, I think they'd get pretty much the same answer if they just compared zonal average (perhaps using equivalent latitude). I'm not suggesting they change anything, just letting them know what I think.
page 14, line 22: I know what equivalent latitude is and I still couldn't understand what the authors were trying to say. Please clean up this definition and make it understandable. Also, is the equivalent latitude calculated independently at each altitude? (in other words, could measurements at different altitudes from a single profile have different eq. latitudes?)

Fig. 10: I don't see a big difference between geographic latitude and equivalent latitude in Fig. 10. The real advantage of equivalent latitude is not in the zone average, where excursions tend to cancel, but the fact that there is much lower scatter among measurements made at a particular equivalent latitude then at the same geographic latitude. The authors might want to find a better way to demonstrate the benefits of eq. latitude.

page 19, line 5-6: 82 hPa is not in the lowermost stratosphere.

---

## Referee Comment (RC2) · Anonymous Referee #2 · 19 Jun 2016

**GENERAL COMMENTS**

This is a well-written paper that describes in detail the SWOOSH database. It documents all source data sets for SWOOSH as well as how the raw data from the individual satellite-based instruments are combined into a variety of zonal mean monthly mean products. The paper provides the appropriate documentation for the SWOOSH database that is likely to be used by a wide range of researchers in stratospheric dynamics, chemistry, and radiative transfer. I think that it needs to be made clearer in the abstract whether SWOOSH is (a) a collation of satellite-based measurements from multiple instruments in one big database, or (b) a merged data product (comprising zonal mean monthly mean values), or (c) both. I have made a few specific comments

below which the authors should address before the papers is accepted for publication. However these are generally minor comments and it should not be a significant amount of work for the authors to bring this paper to a state where can be published in ESSD.

SPECIFIC COMMENTS

Page 2, line 9: Regarding "homogeneous and accurate". Do you mean homogeneous in space and time or just in time? And by accurate do you mean unbiased or of low random uncertainty?

Page 2, line 20: Are changes in surface UV radiation really considered a "climate impact"?

Page 3, line 23: I would suggest replacing "diurnal variability" with "the diurnal cycle in ozone".

Page 5, line 25: The whole profile is excluded, not just the values between 30 and 50 km?

Page 6, line 15: Do comparisons of the ozone product retrieved from the 183 GHz channel and the product retrieved from the 205 GHz channel provide useful information on the uncertainties on 205 GHz ozone data that you are using?

Page 8, line 5: You're talk here about instruments not variables so better to replace "ozone" with "ozonesonde".

Page 14, line 23: This is a very contorted description of equivalent latitude. Why not just say: the equivalent latitude associated with a prescribed PV value is that latitude which encloses the same area as the PV isoline?

Page 14, line 24: I think that you should make it clear that this applies only for long-lived tracers.

Page 14, line 29: How relevant/useful is the equivalent latitude at low geographic latitudes where PV is less representative of the behaviour of a passive tracer? I think

that other people have used a hybrid latitude that is equivalent latitude poleward of 50 degrees and true latitude equator-ward of 30 degrees with a transition zone in between.

Page 17, line 19: I agree with what is written here and it concerns me. Researchers using the equivalent latitude filled geographical data will often end up using monthly mean zonal means that are biased low. Aren't you contaminating your geographical latitude data set by doing this?

Page 17, line 29: I think that this "radial basis function interpolation with an inverse mul-tiquadric function" needs to be explained more thoroughly. Can the actual equations used be provided e.g. in an appendix?

GRAMMAR AND TYPOGRAPHICAL ERRORS

While reading the paper are spotted a few grammatical and typographical errors that I bring to the attention of the authors here should they wish to correct them. In no way should this detract from the excellent quality of the paper.

Page 2, line 3: Replace "1980's" with "1980s". Apostrophes denote contraction or possession and this is neither. Similarly elsewhere in the paper.

Page 5, line 26: Replace "water data" with "water vapour data".

Page 5, line 31: Replace "wit the SAGE" with "with the SAGE".

Page 6, line 11: Either the delete the "approximately" or the $\sim$

Page 6, line 29: I would prefer to see "ground-based measurements" rather than "ground-based data".

Page 7, line 5: Replace "extinctions" with "extinction".

Page 7, line 26: Delete the second "(100 − 1 hPa)".

Page 7, line 27: Replace "is provided" with "are provided".

Page 9, line 30: Replace "to estimated" with "to the estimated".

Page 10, lines 25-26: I don't know what is meant by "the statistical test does not account for seven".

Table 3: There is something anomalous with the "Period" entry for Samoa (looks like there are two sets of values).

Page 12, line 2: Replace "is gridded" with "are gridded".

Page 12, line 11: Replace "of "noise"." as "of as "noise"."

Page 12, line 17: Replace "measurements are used" with "are used".

Page 13, line 2: Replace "per decade" with "per decade in pressure".

Figure 8 caption: Replace "RMMS" with "RMSS"

Page 13, line 24: Replace "magnitude offsets" with "magnitude of the offsets".

Page 14, line 17: Replace "on to" with "onto".

Page 15, line 1: Replace "gridded data is" with "gridded data are".

Page 17, line 15: Replace "samples $\pm 82°$ latitude" with "samples between 82°S and 82°N".

Page 18, line 21: Replace "clearly captures" with "clearly capture".

Page 20, line 16: Replace "data is saved" with "data are saved".

Page 20, line 27: Replace "new data is" with "new data are".

Page 21, line 4: Replace "input in" with "input to".

Page 22, line 4: Replace "algorithm removing" with "algorithm for removing".

---

## Referee Comment (RC3) · Anonymous Referee #3 · 20 Jul 2016

This manuscript introduces a new merged ozone and water vapor data set from satellite limb sounders – SWOOSH – which constitutes a very valuable resource for the study of stratospheric climate variability and change that can be anticipated to be widely used in the community. While the paper is generally well written I miss some key information that I would expect in the documentation of a new database of this importance (these are, to summarize the more detailed comments below, the validation of the dataset and its consistency with other measurement systems, clearer guidance on uncertainty measures), as well as providing better information of how SWOOSH distinguishes it-self from e.g. the GOZCARDS (Froidevaux et al., 2015) or Bodeker scientific BDBP (Bodeker et al., 2013) datasets that are already available. I acknowledge that the paper

states its main goal is to introduce the methodology of merging, but without validation (or at least a rough sanity check) of the product, knowledge on the methodology is not very useful since the reader cannot judge the validity of the methodology applied. I hence suggest some major improvements as detailed below before I can recommend publication, which hopefully will help make the manuscript more valuable to the data user.

**Major comments**

The main problems I see with this manuscript are:

- The authors neglect to put their new database into context with already existing stratospheric databases. I don't agree with the statement in the introduction that 'Despite their chemical and radiative importance in the stratosphere, there have been relatively few attempts at constructing long-term data records of O3 and WV based on vertically resolved satellite limb-based observations of these species'. Given the limited number of satellite instruments covering the time period 1979-2003, how many times can these be merged to the newer satellite instruments (starting 2002 or later) with adding value to previously merged datasets? Providing information on improvements or differences in the merging approaches used is hence a necessity in order not to confuse the data user on which data product should he/she use, but mostly lacking in this document. An obvious omission I see is not citing Bodeker et al. (2013), but then there are other merged datasets that are not mentioned (ESA CCI, Sofieva et al., 2013), wrongly cited (Froidevaux et al., 2015; Hegglin et al., 2014), or just mentioned in passing (Randel and Fu, 2007).

- The manuscript should provide a much better guideline for the end user for what the dataset can be used for and for what not. The information is sprinkled throughout the manuscript, but never brought to a conclusion or into a summary. The inexperienced data user will then just go ahead and use the merged anomaly-filled dataset that is easiest to use (since there are no holes in the record), but which arguably has the largest

uncertainties in reproducing the real atmosphere given the construction methodology. As a more concrete example, the discussion in Appendix A on the investigation of the low bias in Aura-MLS is not drawn upon in the main part of the manuscript and leaves the reader with the open question on how this influences the validity of SWOOSH below 100 hPa. Here again I can't find too useful guidance given by the authors to the user.

- Along, with the previous comment, a more accurate communication of the uncertainties of the database is necessary to prevent false conclusions in studies that e.g. look at trends in these species or compare chemistry-climate models to these observations. The different definitions of uncertainty you provide confused me, not last due to the fact that the labeling of the different sigmas in the figures is not always consistent with what you write in the text. Instead of burying the discussion and definitions of the uncertainties in the Appendix, I would expect this to be a key part of the discussion within the paper and explained in a clearer more concise way. For example, you could show in addition to the applied offsets shown in figures 5 and 6, similar latitude-height distributions of the RMSS or combined standard deviations (possibly for two different time periods in the early/later part of the record). This should more clearly illustrate where the user can trust the dataset and where not (e.g. < 100hPa).

- Another major problem I have is with the communication of the data product that uses anomaly filling of data where there is basically no information from the satellite instruments available. These locations should really get a special uncertainty associated with or be flagged in an obvious way. This information is buried in the text and not brought up in the conclusions anymore where also shortcomings of the database should be summarised and highlighted.

- The manuscript states several times that the evaluations presented would illustrate the use of SWOOSH for studies of variability on different timescales. However, Section 5 provides very limited evidence for this and in my eyes is unsatisfactory if the features shown are not validated against independent instruments. I would have expected a better sanity check of the merged product using long-term observations from other measurement platforms for comparison. The authors could as a suggestion compare their new water vapor product to the Boulder FPH dataset, which would span the whole time period of SWOOSH and hence could be used to check whether the merging worked satisfactorily. Similarly, long ozonesonde records exist (see e.g. Randel and Thompson, 2010), which could be similarly used to test the QBO in the tropics. This is in particular important since the QBO is seen to have very different structures in different instruments (due to sampling, vertical resolution, etc issues see Tegtmeier et al., 2013).

- Finally, the dataset is provided in different coordinate systems and resolutions and I would assume that the uncertainty estimates should increase when moving from 10 down to 2.5 degrees latitude resolution. Also, I would expect that the higher resolution dataset would be noisier than the lower resolution one especially in the early part of the record. It would therefore be important to illustrate and discuss these characteristics and differences, and I suggest to add some figures with meridional profiles of both absolute values and uncertainties to the manuscript.

**Minor and technical comments**

Abstract: The abstract does not provide enough information on the characteristics of the database. I suggest to add that you have 3D and 2D climatologies, different coordinate systems (list them), height range covered (300, 100 hPa upwards to 1, 0.1, 0.001 hPa?), latitude/longitude resolutions available, and that the instruments considered are all only from NASA satellites (in contrast to e.g. Froidevaux et al. 2013 who use also Canadian Space Agency observations).

P1L13 Suggest to delete 'its use for studies of' I don't think you have shown the use of SWOOSH for the study of climate variability in this paper. This is because you haven't compared to other measurements or models.

P1L19 'climate impacts' is a misleading choice of words.

P1L25 see major comment above. I don't think you do the past work on merging datasets of the stratospheric community justice by claiming this.

P2L5 It is confusing to mingle merged total column data sets and vertically resolved data sets here.

P2L8 To my knowledge the SPARC Data Initiative does not offer a merged water vapour data set. Hegglin et al. (2014), see full citation below, should be used instead.

P2L21 A more recent comparison of these satellite instruments supporting this conclusion is given by Tegtmeier et al. (2013).

P2L25 The Kley et al. (2000) reference is outdated given that you would also like to compare Aura-MLS (from 2004 onwards) with these earlier satellite observations. Here a reference to the SPARC Data Initiative water vapour assessment (Hegglin et al., 2013) that you used earlier in the wrong context should be added. This paper shows that Aura-MLS, SAGE II, and HALOE do in fact agree much better within 10

P6L26 This implies that HALOE offers measurements down to the ground, while HALOE data providers now recommend that the data should not be used below 100 hPa. Please amend.

P6L29 Add reference to Hegglin et al. (2013).

P8L26 The coincident criteria you choose are much looser than what is generally used in validation studies (mostly within 6 hours and 400 km). Have you tried to make them stricter and how would this affect your offset uncertainty estimates? It seems scientifically not correct that by loosening your coincidence criteria (which should introduce larger biases) let's you achieve improved uncertainties on your error estimates (on the grounds that you have more profiles in the comparison that affect your standard error of the mean).

P10L5 Looking at figure 3 this implies that Aura-MLS has a high bias of up to 7

P10L8-13 I don't understand your argumentation here nor do I agree with it. After all you should assume that the FPH is your truth. By choosing sat+FPH as your reference you are decreasing the percentage bias estimate which seems arbitrary.

P12L3 The term 'to create statistical agreement' is a misleading terminology to use here. If I understand your methodology right you simply bias-correct the mean. What bringing into 'statistical agreement' means is to do a quantile-adjustment, i.e., also correcting for the variance and variability differences in two datasets. Please change wording.

P13L1 Please specify vertical range.

P13L5-8 This would be an example figure that I would like to see evidenced in the appendix, since the question of the impact of averaging kernels on instrument or model-instrument comparisons is always asked by reviewers and users.

P13L12-15 The study by Hegglin et al. (2014) who introduce a new merging technique based on a chemistry-climate model as transfer function seem to reach the conclusion that a simple merge between two datasets such as HALOE and Aura-MLS does lead to a wrong bias-adjustment due to a potential degradation of HALOE and SAGE II observations towards the end of their lifetimes. Another study by Brinkop et al. (2016) provide additional evidence that support the conclusion of this study. The fact that your methodology is based on bias-adjustment during overlap periods of instruments may hence be a problem for the merging and should be mentioned in this manuscript.

P14L31-P15L1 That's not the full story. Equivalent latitude is expected to steepen gradients also in the subtropics and in the tropopause region. Please amend this statement.

P15L23 onwards: please improve consistency of the annotations of these different standard deviations or uncertainties in the text and the figures 8 and 9.

P17L2 This recommendation to the user of your dataset to use the combined standard deviation (s), which seems equivalent to the light grey shading in figures 8 and 9 (please clarify if this interpretation is not correct), seems scientifically not justifiable to me. The figures show that s is basically constant over the whole time period, which does not reflect in any way that there are known uncertainties stemming from the pure facts that HALOE measurements have a much sparser sampling and have been strongly affected during the early 90's by the Pinatubo aerosol.

P18L6-11 An equivalent statement to this one needs to show up in the conclusion sections.

P18L24 See comment above, there are new studies that claim the drop in the merged HALOE-Aura-MLS datasets is overestimated (Hegglin et al., 2014; Brinkop et al., 2016), so this cannot be used as proof that your dataset is showing the right behavior. Please add this caveat.

P20L5 I don't see where you have explained the differences to the Froidevaux or Randel and Wu methodologies further above.

P21L3 There are indeed non-American satellite instruments as well that provide water vapor measurements from space. These are the Canadian ACE-FTS or the Swedish Odin-SMR, both of which are still in space and would be very useful for extending the water vapor record (at least as long as they can keep up in space as is true for Aura-MLS).

Appendix A: It is not clear to me how this information on the Aura-MLS low bias affects the data screening used in your study.

P22L15 delete 'the' between 'kernel to' and 'degrade'

P22L18-24 This seems to me a very surprising result and in my eyes warrants further investigation. Do you really use the full retrieval procedure that is used in an equivalent way for the retrieval of MLS L2 data? The shape of the a-priori profile seems too close to the L2 data profile so that it would be surprising that other measurement aspects

create the strong oscillations instead. Did you discuss this with the MLS folks?

P23L1 The dry bias in Aura-MLS at this altitude has also been pointed out in the study by Read et al. (2007) and Hegglin et al. (2013).

P23L29 This seems to contradict your interpretation of your own evaluation in P22L18-24.

References: Please use the correct reference (see below) for referring to GOZCARDS, the paper has been published in ACP last year already.

Figure 4: Suggest adding a relative difference plot to make this figure consistent with Figure 3. It is not good practice to cover up the axis tick marks with the legend (in both Figures 3 and 4).

Figure 10: This is not a good color scale to use. It makes HALOE appear to have an artifact in equivalent latitude with a distinct high-bias in the Southern hemisphere middle stratosphere when compared to the other panels. Or is it possibly an artifact of the use of equivalent latitude, which introduces too high values at these latitudes? Please test and comment.

References

Bodeker, G. E., et al., A vertically resolved, global, gap-free ozone database for assessing or constraining global climate model simulations, Earth Syst. Sci. Data, 5, 31–43, doi:10.5194/essd-5-31-2013, 2013.

Brinkop, S., et al., The millennium water vapour drop in chemistry–climate model simulations, Atmos. Chem. Phys., doi:10.5194/acp-16-8125-2016, 2016.

Froidevaux, L., et al., Global OZone Chemistry And Related trace gas Data records for the Stratosphere (GOZCARDS): methodology and sample results with a focus on HCl, H2O, and O3, Atmos. Chem. Phys., doi:10.5194/acp-15-10471-2015, 2015.

Hegglin, M. I., et al., Variation of stratospheric water vapour trends with altitude from

merged satellite data, Nature Geoscience, doi:10.1038/NGEO2236, 2014.

Read, W. G., et al., Aura Microwave Limb Sounder upper tropospheric and lower stratospheric H2O and relative humidity with respect to ice validation, J. Geophys. Res., 112, D24S35, doi:10.1029/2007JD008752, 2007.
* * *

---

## Author Comment (AC1) · 2 Sep 2016

**Review #1 (Prof. Andrew Dessler)**

This paper describes a combined water vapor and ozone data set. This could be a widely used data set and the paper describes it well. I recommend publication after the authors consider my comments below.

My most important comment has to do with the construction of the combined anomaly data. Looking at earlier versions of SWOOSH, I remember that the different satellites had different shape seasonal cycles (at least in certain places), and this caused problems in the combined anomaly data. So I ended up doing my own merge by taking the anomaly time series for each individual satellite and merging those, which required subtracting the offset during their overlap periods. I presume this problem with the seasonal cycles is still an issue. I'm not recommending they change anything in how the data set is presently constructed, but I would suggest that the authors mention something about this as well as the fact that users of the data might want to create their own merged data set.

*We thank Prof. Dessler for the comment on this, as it is possible that SWOOSH users may wish to construct or explore alternative merging methodologies to that which we've provided as the standard SWOOSH product. We have added text in section 4.3 to address this. As we note in the text, because we provide statistics such as the means and N (the number of profiles in a bin) and the individual anomalies, users are free to construct alternative merges with the SWOOSH data set.*

page 7, line 8: The conserved quantity in HALOE is between 1.7 and 1.9 ppmv H2O/ppmv CH4, as described by Dessler and Kim [Determination of the amount of water vapor entering the stratosphere based on HALOE data, J. Geophys. Res., 104, 30,605-30,607, 1999; see Table 1]. I don't know if that'll make a difference, but the authors should use the correct value.

*We reproduced our plot using different slopes, and the values are completely insensitive to the choice of slope. We've changed the value used in the paper to be 1.8, which represents a "middle of the road" value given by Dessler and Kim.*

Also, does "discard the profile below 15 hPa" mean altitudes below 15 hPa or pressures below 15 hPa?

*It means at altitudes below 15 hPa. We've clarified this in the text.*

page 10, line 5: In the calculation of the uncertainty, do the authors take autocorrelation into account? Because of temporal autocorrelation, the number of independent pieces of information is less than the number of months, and this needs to be accounted for. See, e.g., Santer et al. (2000), (Santer et al., 2000), J. Geophys. Res., 105, 7337- 7356, doi: 10.1029/1999jd901105.

*We thank the reviewer for pointing this out. We've tested the impact of adjusting our N based on the lag-1 autocorrelation, as in the Santer et al. paper. Given that the frostpoint data are taken approximately monthly, the autocorrelations are fairly small, between ~ 0.05 and 0.15. An autocorrelation of 0.15 corresponds to a 25% reduction in the degrees of freedom and a 1/SQRT(0.75)= ~%15 inflation in the size of the error bars in our Figures 3 and 4. While this is not particularly large, we have nevertheless included an accounting for autocorrelation in the text and figures now.*

page 10, line 26: Something is wrong with this sentence; the word "seven" doesn't make sense.

*Fixed.*

An aside: the authors spend a lot of effort matching satellite measurements with other satellites and sondes. These approaches have been done for decades, and I find them pretty unconvincing. The "match" criteria is set to get a large number of matches rather than because the criteria actually designate similar air masses. While I don't doubt the answers that the authors get are right, I think they'd get pretty much the same answer if they just compared zonal average (perhaps using equivalent latitude). I'm not suggesting they change anything, just letting them know what I think.

*We agree and expect that the results would be roughly similar as a zonal average. But as we have considered a number of sounding sites across the globe, some locations are in regions of strong (equivalent) latitudinal gradients in tracer mixing ratio (particularly ozone), so including an equivalent latitude constraint on the match provides some improvement in the matching. Also, as we only consider one satellite profile for each ozone/frostpoint sounding, the eq. lat matching actually allows us an objective way to pick one profile from among the potentially large number of profiles that meet the more coarse time and geographic distance criteria.*

page 14, line 22: I know what equivalent latitude is and I still couldn't understand what the authors were trying to say. Please clean up this definition and make it understand- able. Also, is the equivalent latitude calculated independently at each altitude? (in other words, could measurements at different altitudes from a single profile have different eq. latitudes?)

*We have simplified the definition, and clarified that the equivalent latitude is calculated at each altitude of the satellite profile.*

Fig. 10: I don't see a big difference between geographic latitude and equivalent latitude in Fig. 10. The real advantage of equivalent latitude is not in the zone average, where excursions tend to cancel, but the fact that there is much lower scatter among measurements made at a particular equivalent latitude then at the same geographic

latitude. The authors might want to find a better way to demonstrate the benefits of eq. latitude.

*We've changed the figure to better illustrate the benefits of equivalent latitude, particularly with respect to identifying the low ozone concentrations associated with the Antarctic ozone hole. Changes to the figure include changes to the altitude, colorbar type, and color scale, as well as making the figure a contour plot to allow for an easier quantitative comparison. We also removed the SAGE-II data, as the SAGE-II sampling during the chosen month was at too low of a latitude to illustrate the utility of equivalent latitude.*

page 19, line 5-6: 82 hPa is not in the lowermost stratosphere.

*We've fixed the wording to say "lower" stratosphere.*

---

## Author Comment (AC2) · 2 Sep 2016

**Review #2.**

GENERAL COMMENTS

This is a well-written paper that describes in detail the SWOOSH database. It documents all source data sets for SWOOSH as well as how the raw data from the individual satellite-based instruments are combined into a variety of zonal mean monthly mean products. The paper provides the appropriate documentation for the SWOOSH database that is likely to be used by a wide range of researchers in stratospheric dy- namics, chemistry, and radiative transfer. I think that it needs to be made clearer in the abstract whether SWOOSH is (a) a collation of satellite-based measurements from multiple instruments in one big database, or (b) a merged data product (comprising zonal mean monthly mean values), or (c) both. I have made a few specific comments below which the authors should address before the papers is accepted for publication. However these are generally minor comments and it should not be a significant amount of work for the authors to bring this paper to a state where can be published in ESSD.

*We thank the reviewer for their general comments. SWOOSH contains both the data records from individual satellites, as well as a merged product, as we state in the abstract – "SWOOSH includes both individual satellite source data as well as a merged data product."*

SPECIFIC COMMENTS

Page 2, line 9: Regarding "homogeneous and accurate". Do you mean homogeneous in space and time or just in time? And by accurate do you mean unbiased or of low random uncertainty?

*By homogeneous we mean free of artificial jumps in time, and by accurate we mean unbiased. We have changed the wording in the manuscript to be clearer in regards to this point.*

Page 2, line 20: Are changes in surface UV radiation really considered a "climate impact"?

*For clarity we removed the word "climate" in this sentence.*

Page 3, line 23: I would suggest replacing "diurnal variability" with "the diurnal cycle in ozone".

*Fixed*

Page 5, line 25: The whole profile is excluded, not just the values between 30 and 50 km?

*Yes, as per the recommendations of Wang et al (2002). We added the word "entirely" to make clear that the entire profile is removed if this criterium is met.*

Page 6, line 15: Do comparisons of the ozone product retrieved from the 183 GHz channel and the product retrieved from the 205 GHz channel provide useful information on the uncertainties on 205 GHz ozone data that you are using?

*We are unaware of any studies specifically comparing the two. The Livesey et al 2003 paper we cite contains an independent comparison to ozonesondes that forms the basis for their recommendation of using the 205 GHz product.*

Page 8, line 5: You're talk here about instruments not variables so better to replace "ozone" with "ozonesonde".

*Page 8, line 5 reads "**3 In situ balloon measurements vs. satellite observations: Choosing a reference satellite measurement** ", so we don't understand this comment.*

Page 14, line 23: This is a very contorted description of equivalent latitude. Why not just say: the equivalent latitude associated with a prescribed PV value is that latitude which encloses the same area as the PV isoline?

*We have clarified the definition in the text.*

Page 14, line 24: I think that you should make it clear that this applies only for long-lived tracers.

*Done*

Page 14, line 29: How relevant/useful is the equivalent latitude at low geographic latitudes where PV is less representative of the behaviour of a passive tracer? I think that other people have used a hybrid latitude that is equivalent latitude poleward of 50 degrees and true latitude equator-ward of 30 degrees with a transition zone in between.

*The reviewer brings up a good point that at low geographic latitudes PV-based equivalent latitude is not particularly useful. We have added a warning in the last paragraph of section 4.2.*

Page 17, line 19: I agree with what is written here and it concerns me. Researchers using the equivalent latitude filled geographical data will often end up using monthly mean zonal means that are biased low. Aren't you contaminating your geographical latitude data set by doing this?

*We have altered this sentence to read as a warning to users. However, as the "equivalent latitude filled" products we provide are in addition to and separate from the regular geographically gridded products, we don't agree that we are "contaminating" the geographical latitude data set. If users don't want to use the "equivalent latitude filled" products, they can simply use the regular geographic gridded products.*

Page 17, line 29: I think that this "radial basis function interpolation with an inverse multiquadric function" needs to be explained more thoroughly. Can the actual equations used be provided e.g. in an appendix?

*A review of this type of interpolation is beyond the scope of this paper. We've added a reference to the Hardy review of multiquadric functions, and have documented our implementation of the interpolation using IDL.*

GRAMMAR AND TYPOGRAPHICAL ERRORS

While reading the paper are spotted a few grammatical and typographical errors that I bring to the attention of the authors here should they wish to correct them. In no way should this detract from the excellent quality of the paper.

*We thank the reviewer for the very careful review and for pointing out these typos and grammatical issues.*

Page 2, line 3: Replace "1980's" with "1980s". Apostrophes denote contraction or possession and this is neither. Similarly elsewhere in the paper.

*Fixed.*

Page 5, line 26: Replace "water data" with "water vapour data".

*Done*

Page 5, line 31: Replace "wit the SAGE" with "with the SAGE".

*Done*

Page 6, line 11: Either the delete the "approximately" or the ~

*Done*

Page 6, line 29: I would prefer to see "ground-based measurements" rather than "ground-based data".

*Done*

Page 7, line 5: Replace "extinctions" with "extinction".

*Done*

Page 7, line 26: Delete the second "(100 – 1 hPa)".

*Done*

Page 7, line 27: Replace "is provided" with "are provided".

*Done*

Page 9, line 30: Replace "to estimated" with "to the estimated".

*Done*

Page 10, lines 25-26: I don't know what is meant by "the statistical test does not account for seven".

*This has been fixed.*

Table 3: There is something anomalous with the "Period" entry for Samoa (looks like there are two sets of values).

*Fixed*

Page 12, line 2: Replace "is gridded" with "are gridded".

*Fixed*

Page 12, line 11: Replace "of "noise"." as "of as "noise".."

*Done*

Page 12, line 17: Replace "measurements are used" with "are used".

*Done*

Page 13, line 2: Replace "per decade" with "per decade in pressure".

*Done*

Figure 8 caption: Replace "RMMS" with "RMSS"

*Done*

Page 13, line 24: Replace "magnitude offsets" with "magnitude of the offsets".

*Done*

Page 14, line 17: Replace "on to" with "onto".

*Done*

Page 15, line 1: Replace "gridded data is" with "gridded data are".

*Done*

Page 17, line 15: Replace "samples ±82◦ latitude" with "samples between 82◦S and 82◦ N".

*Done*

Page 18, line 21: Replace "clearly captures" with "clearly capture".

*Done*

Page 20, line 16: Replace "data is saved" with "data are saved".

*Done*

Page 20, line 27: Replace "new data is" with "new data are".

*Done*

Page 21, line 4: Replace "input in" with "input to".

*Done*

Page 22, line 4: Replace "algorithm removing" with "algorithm for removing".

*Done*

---

## Author Comment (AC3) · 2 Sep 2016

**Reviewer #3**

This manuscript introduces a new merged ozone and water vapor data set from satellite limb sounders – SWOOSH – which constitutes a very valuable resource for the study of stratospheric climate variability and change that can be anticipated to be widely used in the community. While the paper is generally well written I miss some key information that I would expect in the documentation of a new database of this importance (these are, to summarize the more detailed comments below, the validation of the dataset and its consistency with other measurement systems, clearer guidance on uncertainty measures), as well as providing better information of how SWOOSH distinguishes itself from e.g. the GOZCARDS (Froidevaux et al., 2015) or Bodeker scientific BDBP (Bodeker et al., 2013) datasets that are already available. I acknowledge that the paper states its main goal is to introduce the methodology of merging, but without validation (or at least a rough sanity check) of the product, knowledge on the methodology is not very useful since the reader cannot judge the validity of the methodology applied. I hence suggest some major improvements as detailed below before I can recommend publication, which hopefully will help make the manuscript more valuable to the data user.

**Major comments**

The main problems I see with this manuscript are:

- The authors neglect to put their new database into context with already existing stratospheric databases. I don't agree with the statement in the introduction that 'Despite their chemical and radiative importance in the stratosphere, there have been relatively few attempts at constructing long-term data records of O3 and WV based on vertically resolved satellite limb-based observations of these species'. Given the limited number of satellite instruments covering the time period 1979-2003, how many times can these be merged to the newer satellite instruments (starting 2002 or later) with adding value to previously merged datasets? Providing information on improvements or differences in the merging approaches used is hence a necessity in order not to confuse the data user on which data product should he/she use, but mostly lacking in this document. An obvious omission I see is not citing Bodeker et al. (2013), but then there are other merged datasets that are not mentioned (ESA CCI, Sofieva et al., 2013), wrongly cited (Froidevaux et al., 2015; Hegglin et al., 2014), or just mentioned in passing (Randel and Fu, 2007).

*We thank the reviewer for the suggestions on improving the discussion of the SWOOSH record within the context of the previous/existing efforts at merged data sets.*

*We have clarified the language in the introduction section to better explain the context of SWOOSH and to make clear we are talking about a specific class of data sets that meet the criteria of being vertically resolved (i.e., not total column ozone data sets), timeseries (i.e., and not a climatology like the SPARC Data Initiative data sets by Hegglin and Tegtmeier), and gridded. We also make more clear upfront how the SWOOSH methodology is different than previous efforts.*

*Also, regarding the reference to Bodeker et al 2013 -- The lack of citation to Bodeker et al. 2013 was a mistake. We had mistakenly cited Hassler et al. 2008 as the reference to the BDBP. The Hassler et al. 2008 work underpins the Bodeker et al 2013 paper, but the*

*Bodeker et al. paper describes the merged/gridded product, and is thus the appropriate paper to reference in this section.*

- The manuscript should provide a much better guideline for the end user for what the dataset can be used for and for what not. The information is sprinkled throughout the manuscript, but never brought to a conclusion or into a summary. The inexperienced data user will then just go ahead and use the merged anomaly-filled dataset that is easiest to use (since there are no holes in the record), but which arguably has the largest uncertainties in reproducing the real atmosphere given the construction methodology.

*We agree and thank the reviewer for this idea. We have added a specific set of recommendations in the conclusions section that includes guidance in particular on the use of the merged anomaly-filled data set.*

As a more concrete example, the discussion in Appendix A on the investigation of the low bias in Aura-MLS is not drawn upon in the main part of the manuscript and leaves the reader with the open question on how this influences the validity of SWOOSH below 100 hPa. Here again I can't find too useful guidance given by the authors to the user.

*The screening described in Appendix A has been applied to the Aura MLS data in SWOOSH to remove the affected data, thus there is no need for the user to do anything. We have modified the text at the end of section 2.4 and in Appendix A to make this clear to the reader.*

- Along, with the previous comment, a more accurate communication of the uncertainties of the database is necessary to prevent false conclusions in studies that e.g. look at trends in these species or compare chemistry-climate models to these observations. The different definitions of uncertainty you provide confused me, not last due to the fact that the labeling of the different sigmas in the figures is not always consistent with what you write in the text.

*We have improved the uncertainty discussion (also in response to the next reviewer comment), and have double-checked the notation used in the figures and text to ensure that we've used the notation and terminology consistently.*

Instead of burying the discussion and definitions of the uncertainties in the Appendix, I would expect this to be a key part of the discussion within the paper and explained in a clearer more concise way.

*We believe it is appropriate to keep the detailed derivation of the uncertainties in the Appendix, but we have added additional uncertainty discussion in the manuscript text in order to give the more casual reader an overview of the most important points about the SWOOSH uncertainties.*

For example, you could show in addition to the applied offsets shown in figures 5 and 6, similar latitude-height distributions of the RMSS or combined standard deviations (possibly for two different time periods in the early/later part of the record). This should more clearly illustrate where the user can trust the dataset and where not (e.g. < 100hPa).

*There are always more plots and pieces of information that could be added to a lengthy paper such as this, but we don't feel these plots would add enough value justify their inclusion in the paper. Instead, we've added discussion on where the data set can be "trusted" and where not into the Discussion section.*

- Another major problem I have is with the communication of the data product that uses anomaly filling of data where there is basically no information from the satellite instruments available. These locations should really get a special uncertainty associated with or be flagged in an obvious way. This information is buried in the text and not brought up in the conclusions anymore where also shortcomings of the database should be summarised and highlighted.

*We have added a warning about the anomaly filled data to the Discussion section at the end of the paper. Also, SWOOSH users can easily tell where the data have been filled, either by directly comparing the anomaly-filled arrays to the corresponding non-filled version, or by considering the "N" arrays (arrays containing number of data points that went into the bin). We have added text stating this clearly in the discussion section.*

- The manuscript states several times that the evaluations presented would illustrate the use of SWOOSH for studies of variability on different timescales. However, Section 5 provides very limited evidence for this and in my eyes is unsatisfactory if the features shown are not validated against independent instruments. I would have expected a better sanity check of the merged product using long-term observations from other measurement platforms for comparison. The authors could as a suggestion compare their new water vapor product to the Boulder FPH dataset, which would span the whole time period of SWOOSH and he nce could be used to check whether the merging worked satisfactorily. Similarly, long ozonesonde records exist (see e.g. Randel and Thompson, 2010), which could be similarly used to test the QBO in the tropics. This is in particular important since the QBO is seen to have very different structures in different instruments (due to sampling, vertical resolution, etc issues see Tegtmeier et al., 2013).

*To the extent possible, validation of SWOOSH against independent measurements is a laudable task. Such a "validation" of SWOOSH will be the focus of future work and is beyond the scope of this already quite lengthy paper, as it is primarily intended to document the methodology used.*

*In regards to ozone, SWOOSH has already participated in several 'validation' activities as part of the SI2N project. These validation activities included comparisons to independent ground-based measurements, independent satellite data sets, and other*

*merged data sets. The results are summarized in several papers in the peer-reviewed literature (Harris et al., 2015; Tummon et al., 2015; Hubert et al., 2015). It was an oversight on our part to not discuss and reference these papers here, so we've added discussion and references in Section 5. In that section we've also added a timeseries comparison to the tropical ozonesonde site at Natal that was included in our ozonesondes database to further expand on the discussion of QBO-related variability.*

*With regards to water vapor validation, we have added a comparison between SWOOSH and the Boulder record and a discussion of the results. A number of studies have addressed the divergence between the Boulder FP record and the satellite records; exact reasons for those differences are still not well understood. Given that the purpose of SWOOSH is to reproduce the short and long-term variations in the underlying source records, it comes as no surprise that there are differences.*

- Finally, the dataset is provided in different coordinate systems and resolutions and I would assume that the uncertainty estimates should increase when moving from 10 down to 2.5 degrees latitude resolution. Also, I would expect that the higher resolution dataset would be noisier than the lower resolution one especially in the early part of the record. It would therefore be important to illustrate and discuss these characteristics and differences, and I suggest to add some figures with meridional profiles of both absolute values and uncertainties to the manuscript.

*Yes, the uncertainty estimates are different for each gridding, as they are calculated independently for each grid (i.e., the 10 degree grid is not just the average of the two 5 degree bins). This is now stated in Section 4.2. We don't feel it is necessary to add figures to the paper to illustrate this.*

**Minor and technical comments**

Abstract: The abstract does not provide enough information on the characteristics of the database. I suggest to add that you have 3D and 2D climatologies, different coordinate systems (list them), height range covered (300, 100 hPa upwards to 1, 0.1, 0.001 hPa?), latitude/longitude resolutions available, and that the instruments considered are all only from NASA satellites (in contrast to e.g. Froidevaux et al. 2013 who use also Canadian Space Agency observations).

*Done*

P1L13 Suggest to delete 'its use for studies of' I don't think you have shown the use of SWOOSH for the study of climate variability in this paper. This is because you haven't compared to other measurements or models.

*Given that we've shown examples of the tape recorder and QBO ozone variability in SWOOSH, and have now included a section on comparison to independent observations, we have left this line in.*

P1L19 'climate impacts' is a misleading choice of words.

*This has been changed.*

P1L25 see major comment above. I don't think you do the past work on merging datasets of the stratospheric community justice by claiming this.

*See response to major comment above.*

P2L5 It is confusing to mingle merged total column data sets and vertically resolved data sets here.

*The point of this sentence was to contrast SWOOSH with the total column data sets.*

P2L8 To my knowledge the SPARC Data Initiative does not offer a merged water vapour data set. Hegglin et al. (2014), see full citation below, should be used instead.

*Done.*

P2L21 A more recent comparison of these satellite instruments supporting this conclusion is given by Tegtmeier et al. (2013).

*This reference has been added.*

P2L25 The Kley et al. (2000) reference is outdated given that you would also like to compare Aura-MLS (from 2004 onwards) with these earlier satellite observations. Here a reference to the SPARC Data Initiative water vapour assessment (Hegglin et al., 2013) that you used earlier in the wrong context should be added. This paper shows that Aura-MLS, SAGE II, and HALOE do in fact agree much better within 10%

*Sentence edited and reference added.*

P6L26 This implies that HALOE offers measurements down to the ground, while HALOE data providers now recommend that the data should not be used below 100 hPa. Please amend.

*Clarification added.*

P6L29 Add reference to Hegglin et al. (2013).

*Done*

P8L26 The coincident criteria you choose are much looser than what is generally used in validation studies (mostly within 6 hours and 400 km). Have you tried to make them stricter and how would this affect your offset uncertainty estimates? It seems scientifically not correct that by loosening your coincidence criteria (which should introduce larger biases) let's you achieve improved uncertainties on your error estimates (on the grounds that you have more profiles in the comparison that affect your standard error of the mean).

*We've chosen a set of criteria that are balance between trying to maximize the quantity of matches and ensuring the matches are of high quality. We have thoroughly explored the phase space of time/distance/eq. lat matching, and have found our criteria to perform best. Although our time/distance matching criteria is looser than some other studies, the use of an equivalent latitude matching criteria more than makes up for this looseness, and allows us to include satellite matches would otherwise be lost by stricter time/distance criteria. And as noted by Reviewer #1, in the stratosphere the results are unlikely to be highly sensitive to the matching choice.*

P10L5 Looking at figure 3 this implies that Aura-MLS has a high bias of up to 7

*This comment apparently got truncated; we do not know what the reviewer is trying to say.*

P10L8-13 I don't understand your argumentation here nor do I agree with it. After all you should assume that the FPH is your truth. By choosing sat+FPH as your reference you are decreasing the percentage bias estimate which seems arbitrary.

*We have clarified the reasoning in this sentence. Briefly, because WV values are physically constrained to be positive, the distribution of percent differences is non-normal and positively skewed if one uses FPH as the reference. These conditions preclude the use of the t-test for differences between the population means. Below 100 hPa, this effect can be very large, as there is a large dynamic range of WV, and any mismatch in the profiles leads to large positive mean percent differences if one of the instruments is chosen as the reference. Take for example two sets of matched profiles, one where sat=1 and FP = 100, and the other where sat=100 and FP = 1. If we compute the average percent difference using the 'conventional' definition (i.e., (sat-FP)/FP\*100), then the average percent difference is 0.5 \* ( -99/100 + 99/1)\*100 = 4900%. This would imply that the satellite data are ~5000% high biased relative to the FP, when in fact the value should be zero. In the case of using the average between the FP and satellite as the reference, this is indeed the case.*

*We also note that further evidence of the problem with using the 'conventional' definition is given by the fact that the mean and median percent difference are wildly different under this definition, and are almost exactly the same under the definition we've used in the paper.*

P12L3 The term 'to create statistical agreement' is a misleading terminology to use here. If I understand your methodology right you simply bias-correct the mean. What bringing into 'statistical agreement' means is to do a quantile-adjustment, i.e., also correcting for the variance and variability differences in two datasets. Please change wording.

*We changed the wording.*

P13L1 Please specify vertical range.

*Done*

P13L5-8 This would be an example figure that I would like to see evidenced in the appendix, since the question of the impact of averaging kernels on instrument or model-instrument comparisons is always asked by reviewers and users.

*We feel that the existing information regarding the insensitivity of the offsets to vertical averaging assumptions is sufficient, and that adding more content to this paper is unnecessary.*

P13L12-15 The study by Hegglin et al. (2014) who introduce a new merging technique based on a chemistry-climate model as transfer function seem to reach the conclusion that a simple merge between two datasets such as HALOE and Aura-MLS does lead to a wrong bias-adjustment due to a potential degradation of HALOE and SAGE II observations towards the end of their lifetimes. Another study by Brinkop et al. (2016) provide additional evidence that support the conclusion of this study. The fact that your methodology is based on bias-adjustment during overlap periods of instruments may hence be a problem for the merging and should be mentioned in this manuscript.

*We have altered the wording in the manuscript to read "Thus, drifts or other unphysical changes in individual satellite records, if they exist, are not accounted for in SWOOSH".*

P14L31-P15L1 That's not the full story. Equivalent latitude is expected to steepen gradients also in the subtropics and in the tropopause region. Please amend this statement.

*Done.*

P15L23 onwards: please improve consistency of the annotations of these different standard deviations or uncertainties in the text and the figures 8 and 9.

*We checked and did not find any inconsistencies in the notation, and because the reviewer does not give a specific example of an inconsistency this comment is not possible for us to address further.*

P17L2 This recommendation to the user of your dataset to use the combined standard deviation (s), which seems equivalent to the light grey shading in figures 8 and 9 (please clarify if this interpretation is not correct), seems scientifically not justifiable to me. The figures show that s is basically constant over the whole time period, which does not reflect in any way that there are known uncertainties stemming from the pure facts that HALOE measurements have a much sparser sampling and have been strongly affected

during the early 90's by the Pinatubo aerosol.

*As we discussed in the text, the retrieval uncertainties provided by the instrument teams are problematic for water vapor (i.e., likely too large for SAGE-II, and too small for HALOE). For this reason, SWOOSH users may wish to use the standard deviation of the data as a more consistent measure of 'uncertainty' throughout the record. Alternatively, users could use the standard error of the mean, defined as the standard deviation divided by SQRT(N). In the case of the std. error, the sparser sampling in the early 90's pointed out by the reviewer would indeed be reflected in the uncertainty estimate. That said, the magnitude of a standard error in this case becomes vanishingly small in the later period because of the large number of samples by Aura MLS (N typically of 6000 month$^{-1}$ for a 10° latitude band, leading to std. errors ~ 0.003 ppmv for WV in the stratosphere). It is for this reason that we recommended using the combined standard deviation during the pre Aura MLS period.*

*We recognize that different users and different types of analysis may necessitate the use of different metrics of 'uncertainty'. For this reason, we have removed the explicit recommendation to use the standard deviation in the manuscript.*

P18L6-11 An equivalent statement to this one needs to show up in the conclusion sections.

*Done.*

P18L24 See comment above, there are new studies that claim the drop in the merged HALOE-Aura-MLS datasets is overestimated (Hegglin et al., 2014; Brinkop et al., 2016), so this cannot be used as proof that your dataset is showing the right behavior. Please add this caveat.

*The goal of SWOOSH is to faithfully represent the input data sets used, so the fact that the model-based studies cited above may show some disagreements is irrelevant to the statement here. The point of this statement is to show that in comparison to other studies using similar data, SWOOSH gives similar results.*

P20L5 I don't see where you have explained the differences to the Froidevaux or Randel and Wu methodologies further above.

*We have clarified this sentence and added text in the introduction section of the paper to make the differences more clear.*

P21L3 There are indeed non-American satellite instruments as well that provide water vapor measurements from space. These are the Canadian ACE-FTS or the Swedish Odin-SMR, both of which are still in space and would be very useful for extending the water vapor record (at least as long as they can keep up in space as is true for Aura-MLS).

*We have altered the sentence to remove the claim that Aura MLS is the **only** data set*

*available. Our intention was to state that the Aura MLS is the only **suitable** instrument (i.e., high quality and complete sampling) for the SWOOSH record. In our opinion the ACE-FTS sampling is too poor and the Odin-SMR data quality is too poor to be included in SWOOSH.*

Appendix A: It is not clear to me how this information on the Aura-MLS low bias affects the data screening used in your study.

*As we stated in P8L1-2 of the original manuscript, "Additional filtering of the Aura MLS WV data in the UTLS is described in Appendix A". To try and make more clear that Appendix A describes the algorithm for removing low-biased MLS data in the UTLS, we have changed this sentence to read "Additional filtering of the Aura MLS WV data to remove low-biased data in the UTLS is described in Appendix A". We have also modified the sentence at the end of section 3.1 to read "Additional screening of the Aura MLS data set to remove low-biased WV data in the UTLS is discussed in Appendix A"*

P22L15 delete 'the' between 'kernel to' and 'degrade'

*Done*

P22L18-24 This seems to me a very surprising result and in my eyes warrants further investigation. Do you really use the full retrieval procedure that is used in an equivalent way for the retrieval of MLS L2 data?

*We don't use the "full retrieval procedure" that is used by MLS, as that would involve retrieveing geophysical parameters from the L1 radiances. As we outline in the appendix, we use the procedure described in Read et al. (2007 – note Read is also a coauthor of this paper) that involves convolving the high vertical resolution FP profile with the MLS averaging kernel and a priori profile.*

The shape of the a-priori profile seems too close to the L2 data profile so that it would be surprising that other measurement aspects create the strong oscillations instead. Did you discuss this with the MLS folks?

*One of the MLS science team members (Bill Read) is a coauthor on this paper, and we've discussed this issue in detail not only with him but also other MLS team members (e.g., Nathaniel Livesey, Alyn Lambert)*

P23L1 The dry bias in Aura-MLS at this altitude has also been pointed out in the study by Read et al. (2007) and Hegglin et al. (2013).

*We've cited the previous validation work in the last sentence of Appendix A. The Hegglin et al (2013) reference was not included in the original manuscript, but it has been added in the revised manuscript.*

P23L29 This seems to contradict your interpretation of your own evaluation in P22L18-24.

*As mentioned in our reply above, there is a difference between the averaging kernel/smoothing procedure from Read et al. (2007) and the actual MLS retrieval, so these statements are not contradictory.*

References: Please use the correct reference (see below) for referring to GOZCARDS, the paper has been published in ACP last year already.

*Done*

Figure 4: Suggest adding a relative difference plot to make this figure consistent with

Figure 3.

*Done*

It is not good practice to cover up the axis tick marks with the legend (in both Figures 3 and 4).

*Done*

Figure 10: This is not a good color scale to use. It makes HALOE appear to have an artifact in equivalent latitude with a distinct high-bias in the Southern hemisphere middle stratosphere when compared to the other panels. Or is it possibly an artifact of the use of equivalent latitude, which introduces too high values at these latitudes? Please test and comment.

*We have changed the color scale. See also comment in response to Reviewer #1.*